# The distinct roles of two intertidal foraminiferal species in phytodetrital carbon and nitrogen fluxes - results from laboratory feeding experiments

Julia Wukovits[1], Max Oberrauch[1], Annekatrin J. Enge[1], Petra Heinz[1]

[1] University of Vienna, Department of Palaeontology, Althanstrasse 14, 1090 Vienna, Austria

*Correspondence to*: Julia Wukovits (julia.wukovits@univie.ac.at)

**Abstract.** Benthic foraminifera play a major role as primary consumers and detrivores redistributing organic carbon and nitrogen in intertidal environments. Here we compared the differences of phytodetrital carbon and nitrogen intake and turnover of two dominant intertidal foraminifera, *Ammonia tepida* and *Haynesina germanica*. Their lifestyles in relation to feeding behaviour (feeding preferences, intake and turnover of phytodetrital carbon and nitrogen) and temperature adaptations were compared to obtain a closer definition of their specific roles in intertidal organic matter processing. For this comparison, we carried out a series of short-term laboratory incubations with stable isotope labelled ($^{13}$C & $^{15}$N) detritus as food source. We compared the response of the two species to diatom detritus at three different temperatures (15°C, 20°C, 25°C). *Ammonia tepida* showed a very high, temperature-influenced intake and turnover rates with more excessive carbon turnover, compared to nitrogen. The quite low metabolic nitrogen turnover in *H. germanica* was not affected by temperature and was higher than the carbon turnover. This might be related with the chloroplast husbandry in *H. germanica* and its lower demands for food derived nitrogen sources. *Ammonia tepida* prefers a soft chlorophyte food source over diatom detritus, which is harder to break down. In conclusion, *A. tepida* shows a generalist behaviour that links with high fluxes of organic matter (OM). Due to its high rates of OM processing and abundances, we conclude that *A. tepida* is an important key-player in intertidal carbon and nitrogen turnover, specifically in the short-term processing of OM and the mediation of dissolved nutrients to associated microbes and primary producers. In contrast, *H. germanica* is a highly specialized species with low rates of carbon and nitrogen budgeting.

## 1 Introduction

Benthic foraminifera are ubiquitous marine protists and highly abundant in coastal sediments (Lei et al., 2014; Mojtahid et al., 2016; Murray and Alve, 2000). Coastal sediments represent the largest pool of marine particulate organic matter (OM), despite their rather small area (less than 10% of the ocean floor), and play an essential role in global carbon and nitrogen cycles (Jahnke, 2004). Oceanic and terrestrial systems are connected by the carbon cycling in coastal waters, which contribute to a major part of the global carbon cycles and budgets (Bauer et al., 2013; Cai, 2011; Cole et al., 2007; Regnier et al., 2013). Estuaries are an important source for organic matter in coastal systems and were estimated to account for ~ 40% of oceanic phytoplankton primary productivity (Smith and Hollibaugh, 1993). Most estuarine areas are considered to be net heterotrophic,

or act as carbon sinks, respectively (Caffrey, 2003, 2004; Cai, 2011; Herrmann et al., 2015). In general, 30% of overall coastal carbon is lost by metabolic oxidation (Smith and Hollibaugh, 1993). Foraminifera are highly abundant in estuarine sediments and contribute strongly to these processes (Alve and Murray, 1994; Cesbron et al., 2016; Moodley et al., 2000; Murray and Alve, 2000). They feed on various sources of labile particulate OM, including microalgae and detritus, and provide a pivotal link in marine carbon cycles and food webs (Bradshaw, 1961; Goldstein and Corliss, 1994; Heinz, 2001; Lee et al., 1966; Lee

and Muller, 1973; Nomaki et al., 2005a, 2005b, 2006, 2009, 2011). The nitrogen compounds of OM particles are usually remineralized to ammonium ($NH_4^+$). In this way, nitrogen gets again available as nutrient for primary productivity. A major part of this process is attributed to prokaryotic degraders, but protists are also involved in the process of regeneration of organic nitrogen compounds (Ferrier- Pages and Rassoulzadegan, 1994; Ota and Taniguchi, 2003; Verity et al., 1992). Due to their high abundances, we consider, that foraminifera contribute a large part to this OM reworking and the regeneration of carbon

and nitrogen compounds from particulate OM sources, e.g. phytodetritus. In this study, we quantify the bulk OM-derived carbon and nitrogen release, which originates rather via excretion of organic carbon and nitrogen compounds (vesicular transport of metabolic waste products), respiration or diffusion of inorganic carbon and nitrogen by these single celled microorganisms.

Environmental conditions of temperate tidal flats are physiologically challenging (high fluctuations of physical and chemical

parameters, e.g. temperature and/or OM quality) and therefore host very few, highly adapted foraminifera species. Monospecific or near monospecific foraminiferal communities are characteristic for temperate, estuarine regions (Alve and Murray, 1994, 2001; Hayward, 2014; Martins et al., 2015; Saad and Wade, 2017). *Ammonia tepida* and *Haynesina germanica* are typical representatives of these communities and their standing crop can reach more than 150 individuals per $cm^3$ (Alve and Murray, 2001; Mojtahid et al., 2016; Wukovits et al., 2018). Typically, tidal flats offer a high availability of food sources

for phytodetrivores or herbivores feeding on microalgae. But dense populations of *A. tepida* communities can deplete sediments from OM sources and consequently control benthic meiofaunal community structures (Chandler, 1989). Therefore, resource partitioning or different metabolic strategies can be beneficial for foraminifera which share the same spatial and temporal habitats.

Early experimental investigations and monitoring studies suggest feeding preferences or selective feeding in littoral

foraminifera. However, these studies rely on indirect observations from environmental monitoring (Hohenegger et al., 1989; Papaspyrou et al., 2013) or from a laboratory study focusing on the more diverse saltmarsh communities (Lee and Muller, 1973). The latter study revealed, that foraminiferal salt marsh communities are characterized by highly specialized feeding strategies. Analogically, the close spatial coexistence of *A. tepida* and *H. germanica* might be also based on different feeding strategies and different preferences of other environmental variables. A major, important difference between the two species

subject to this study is the fact, that *H. germanica* hosts functional plastids derived from ingested microalgae (Jauffrais et al., 2016; Lopez, 1979), a phenomenon known as kleptoplasty, which was first described for a sacoglossan opisthobranch (Trench, 1969). It was shown, that diatom-derived chloroplasts in the cytoplasm of *H. germanica* retain their function (as photosynthetically active kleptoplasts) for up to two weeks (Jauffrais et al., 2016). Further, there is recent proof that *H.*

*germanica* takes up inorganic carbon and nitrogen sources ($HCO_3$ and $NH_4^+$) from the surrounding seawater, most likely to generate metabolites in autotrophic-heterotrophic interactions with its kleptoplasts (LeKieffre et al., 2018). Consequently, the mixotrophic lifestyle of *H. germanica* might lead to a lower demand of carbon and nitrogen sources and thus to a lower ingestion of various particulate OM sources as food sources. In contrary, food-derived chloroplasts in *A. tepida* lose their photosynthetic activity after a maximum of 24 hours (Jauffrais et al., 2016). Species of the genus *Ammonia* are described to take up significant amounts of microalgae and phytodetritus of different origin. Laboratory feeding experiments have shown, that *A. tepida* responds to several food sources, including different live microalgae (chlorophytes and diatoms) and chlorophyte and diatom detritus (Bradshaw, 1961; LeKieffre et al., 2017; Linshy et al., 2014; Pascal et al., 2008; Wukovits et al., 2017, 2018). Whereas, *H. germanica* shows a low affinity to chloroplast detritus food sources (Wukovits et al., 2017), but feeds actively on diatoms (Ward et al., 2003) and takes up inorganic, dissolved carbon and nitrogen compounds (LeKieffre et al., 2018). Both species are found in muddy coastal sediments containing high loads of nutrients or OM (Armynot du Chatelet et al., 2009; Armynot du Châtelet et al., 2004). But considering their different feeding strategies, both species might play distinct roles in the reworking of OM. Recent literature still lacks direct, quantitative comparisons of foraminiferal species-specific OM-derived C & N ingestion and release. Therefore, this study aims to compare and quantify variations in their respective uptake of OM (phytodetritus).

Temperature has a strong impact on metabolic rates and can therefore play another major role in niche separation or in species-specific adaptations in the consumer community. Benthic foraminifera show strong metabolic responses to temperature fluctuations (Bradshaw, 1961; Cesbron et al., 2016; Heinz et al., 2012). Therefore, seasonal temperature fluctuations and human induced global warming can have a strong impact on foraminiferal community compositions and foraminiferal carbon and nitrogen fluxes. In estuaries e.g. temperature acts in many cases as the most controlling factor on metabolic rates and on net ecosystem metabolism (Caffrey, 2003). To examine the effect of temperature on foraminiferal OM processing, temperature variations were included in our studies. In summary, the aim of this study was to obtain a closer definition of the ecological feeding niches of *A. tepida* and *H. germanica* in relation to intertidal fluxes of OM and OM processing at different temperatures. Additionally, this study offers the first estimates for the release of OM derived carbon and nitrogen in foraminifera. To reach our aim, we carried out laboratory feeding experiments with stable isotope labelled ($^{13}C$ & $^{15}N$) food sources (chlorophyte detritus: *Dunaliella tertiolecta*, diatom detritus: *Phaeodactylum tricornutum*). We compared diatom detritus intake and retention of phytodetrital carbon (pC) and nitrogen (pN) of *A. tepida* and *H. germanica* at three different temperatures (15°C, 20°C, 25°C). The evaluation of the metabolic costs of pC and pN during a 24 hour starvation period can further help to explain species specific OM processing due to metabolic nutrient budgets. Further, both food sources were offered simultaneously to *A. tepida* to identify feeding preferences of this species. Finally, we collected quantitative data of the abundances of both species in the sampling area to estimate species-specific contributions to intertidal fluxes of OM-derived carbon and nitrogen.

## 2 Material and Methods

### 2.1 Sampling area & sample preparation

The sampling area is located at the Elbe river estuary in the German Wadden Sea (Fig. 1). Samples were collected at low tide in April 2016, close to the shoreline. Three sediment cores (4.5 cm diameter) were taken in random spacing within an area of ~ 4 m². The uppermost centimetre of the cores was fixed with a mixture of ethanol and Rose Bengal to stain the cytoplasm of live foraminifera. At the University of Vienna, the sediment core material was sieved to obtain size fractions of 125 – 250 µm, 250 – 355 µm and < 355 µm. Brightly stained (living) foraminifera were identified and counted to calculate abundances (individuals per m²) to estimate the relevance of *A. tepida* and *H. germanica* in intertidal OM fluxes.

For the laboratory experiments, sediment was collected at low tide from the uppermost sediment layer and sieved in the field over 125 µm and 500 µm to remove larger meiofauna and organic components. Sampling trips to collect material for laboratory experiments were done in April 2015 and 2016. The sediment was filled into plastic containers with seawater and transported back to the University of Vienna. The sediment samples were kept within aquaria, containing filtered water collected at the sampling site. Foraminifera were picked from the sediment in sufficient number and collected in crystallizing dishes, containing a layer of North Sea sediment (< 63 µm) and filtered North Sea water (NSW). They were fed with a mixture of live *D. tertiolecta* and *P. tricornutum* once to twice a week until the beginning of the experiments. Live individuals were identified by showing bright and intensive cytoplasm colour, cyst formation (in case of *A. tepida*), material gathered around the aperture, and movement tracks in the sediment. The experiments started after accumulation of sufficient foraminiferal material three weeks after the field sampling.

### 2.2 Production of artificial phytodetritus

Labelled food was produced by growing *D. tertiolecta* and *P. tricornutum* (SAG 1090-1a) in stable isotope-enriched growth medium. Algae were cultured in sterile 5 L Erlenmeyer bottles, containing F1/2 growth medium (Guillard, 1975; Guillard and Ryther, 1962) enriched with aliquots of 98 atom%$NaH^{13}CO_3$ and 98 atom%$Na^{15}NO_3$ (SigmaAldrich). The algae culture medium for Experiment 1 (*P. tricornutum*) was produced with filtered NSW and enriched with 0.6 mM $NaH^{13}CO_3$ and 0.9 mM $NaNO_3$ ($Na^{14}NO_3 : Na^{15}NO_3 \rightarrow 5.25 : 1$), along with the stock solutions for the F/2 standard protocol. The culture medium for *D. tertiolecta* ($^{13}C$ single labelled) in Experiment 2 was produced with filtered NSW, the stock solutions according to the F/2 standard protocol, and additionally enriched with 1.5 mM $NaH^{13}CO_3$ and for *P. tricornutum* ($^{15}N$ single labelled) with 1.5 mM $NaHCO_3$ (natural abundance) and with 0.9 mM $NaNO_3$ ($Na^{14}NO_3 : Na^{15}NO_3 \rightarrow 5.25 : 1$) along with the stock solutions for the F/2 standard protocol. The algae cultures were incubated at 20°C (type ST 2 POL-ECO Aparatura incubation chambers) at a 18 hrs:6 hrs light:dark cycle and bubbled with ambient air. Cultures were harvested at stationary growth (after 14-16 days) by centrifugation, washed three times in sterile, carbon and nitrogen free artificial seawater, shock frozen with liquid nitrogen, and lyophilized to get $^{13}C$ and $^{15}N$-labeled phytodetritus (cf. Wukovits et al., 2017). Three batches of algae were produced.

Final isotopic concentrations were: *P. tricornutum* 7 atom% $^{13}$C and 15 atom% $^{15}$N (Experiment 1), *D. tertiolecta* 22 atom% $^{13}$C (Experiment 2), and *P. tricornutum* 14 atom% $^{15}$N (Experiment 2).

### 2.3 Experiment 1: Nutrient demand and temperature response of *A. tepida* and *H. germanica*

Fifty to fifty five specimens of *A. tepida* and or *H. germanica* respectively, of the size fraction 250 – 355 µm were distributed into separate wells on a 6 well plate, containing NSW (12 mL per well, salinity: 28 PSU, practical salinity units, which lies in the range of our measurements from seawater at the sampling site: 24 – 30 PSU). In total, triplicate samples were prepared. The food source, *P. tricornutum* (1.5 g dry weight m$^{-2}$) was added into each well. Wells were then covered with a headspace to prevent evaporation and were incubated at 15°C, 20°C or 25°C (Table 1).The specimens were incubated at a 12 hrs : 12 hrs

light:dark cycle, starting the incubation with the light cycle. Two equal setups were prepared for incubation. The first setup was terminated after a 24 hour incubation period to determine the intake of *P. tricornutum* detritus per species and temperature ('24 hrs fed'). The experimental period of 24 hours was chosen to avoid potential bacterial activity and to maintain system stability. The specimens were removed from the wells, transferred to Eppendorf© tubes and frozen at -20°C. The specimens of the second setup were washed three times in carbon and nitrogen free artificial seawater after the 24 hour incubation period

and transferred to crystallizing dishes (9 cm diameter), containing 150 mL filtered NSW and covered with parafilm. Subsequently, the dishes were incubated for another 24 hours (15°C, 20°C, 25°C; 12 hours light, 12 hours dark, starting with the light cycle) without food. These samples were analysed to determine the remaining phytodetrital carbon and nitrogen after a 24 hour starvation period ('24 hrs starved').

### 2.4 Experiment 2: Feeding preferences of *A. tepida*

This experiment was carried out at 20°C, since *A. tepida* specimens collected in this area showed a good feeding response at this temperature (Wukovits et al., 2017). *Ammonia tepida* individuals were incubated at 20°C within 6 well plates (55 individuals per triplicate/well, size fraction 250 – 355 µm). Each well was filled with 12 mL NSW. After acclimation of the individuals within the plates, three different dietary setups were established (Table 1). The first diet consisted of chlorophyte derived detritus, uniformly $^{13}$C labelled (*D. tertiolecta*, 1.5 g dry weight cm$^{-2}$), the second was diatom detritus (*P. tricornutum*,

1.5 g dry weight cm$^{-2}$), uniformly $^{15}$N labelled, and the third consisted of a homogenized mixture of both food sources (0.73 g cm$^{-2}$ each). The differential labelling approach allows calculation of nutrient uptake for the distinct phytodetritus source after determination of respective algal carbon and nitrogen composition. Triplicate samples were taken after 1 hour, 3 hours, 6 hours, 12 hours, and 24 hours, and specimens were frozen at -20°C for subsequent isotope ($^{13}$C/$^{12}$C and $^{15}$N/$^{14}$N) and elemental analysis (total organic carbon (TOC) and total nitrogen (TN)). Similarly as in Experiment 1, plates were incubated at a 12 hrs :

12 hrs light:dark cycle, starting the incubation with the light cycle. The algal C:N ratio was used to calculate the pN aliquot for pC of the $^{13}$C labelled chlorophyte and pC for the $^{15}$N labelled diatom food source, for a better visual comparison of the food intake (this serves as a rough estimate of equivalent pC or pN intake at the two diets). This experiment was solely carried

out with *A. tepida*, since the sediment did not contain sufficient individuals of *H. germanica* to set up a parallel run with this species.

## 2.5 Sediment core data and foraminiferal abundances

Sediment core samples (uppermost cm) were sieved to fractionate size classes (125 – 250 µm, 250 – 355 µm, < 355 µm). Rose Bengal-stained individuals were counted for each size fraction to obtain abundance data for the live foraminiferal community at the sampling date. Nutrient budget data from the laboratory experiments (individual TOC, TN, pC, pN), together with the foraminiferal abundances counted from the sediment cores, were used to estimate the range of foraminiferal contributions to sedimentary carbon and nitrogen pools and fluxes. In case of *H. germanica*, these contributions were only estimated for the 250 – 355 µm fraction (as used in laboratory experiments). For *A. tepida*, the 125 – 250 µm fraction was included to the estimation, using size fraction and feeding relationships from Wukovits et al. (2018). Further, the abundances of *A. tepida*, as derived by the latter study, were compared with the recent study.

## 2.6 Sample preparation and isotope analysis

Prior to cytoplasm isotope analysis, foraminifera were carefully cleaned from adhering particles in carbon and nitrogen free artificial seawater, rinsed with ultrapure water in a last cleaning step to remove salts, transferred to tin capsules, and dried at 50°C for several hours. Subsequently, the foraminifera were decalcified with 10 – 15 µL 4 % HCl, and kept at 50°C for three days in a final drying step (Enge et al., 2014, 2016; Wukovits et al., 2017, 2018). The optimum range for isotope and elemental analysis was 0.7 – 1.0 mg cytoplasmic dry weight. In the 250 µm size fraction, 30 – 40 individuals met this criterion. Tools for preparation (hairbrush, needles, tin capsules, tweezers) were rinsed with dichloromethane ($CH_2Cl_2$) and methanol ($CH_4O$) (1:1, v:v). Glassware for microscopy was combusted at 500°C for 5h. The samples were analysed at the Large-Instrument Facility for Advanced Isotope Research at the University of Vienna (SILVER). Ratios of $^{13}C/^{12}C$, $^{15}N/^{14}N$ and the content of organic carbon and nitrogen were analysed with an Isotope Ratio Mass Spectrometer (IRMS; DeltaPLUS, Thermo Finnigan) coupled with an interface (ConFlo III, Thermo Finnigan) to an elemental analyzer (EA 1110, CE Instruments). Isotope ratio data, the Vienna Pee Dee Belemnite standard for C ($R_{VPDB} = 0.0112372$) and the standard for atmospheric nitrogen for N ($R_{atm}N = 0.0036765$) were used to calculate atom% of the samples, where X is $^{13}C$ or $^{15}N$:

$$\text{atom}\%X = \frac{100 \times R_{standard} \times \left(\frac{\delta X_{sample}}{1000} + 1\right)}{1 + R_{standard} \times \left(\frac{\delta X_{sample}}{1000} + 1\right)}, \tag{1}$$

Intake of pC and pN into foraminiferal cytoplasm was calculated by determining the excess ($E$) of isotope content within the samples using natural abundance data and data of enriched samples (Middelburg et al., 2000):

$$E = \frac{(atom\%X_{sample} - atom\%X_{background})}{100}, \tag{2}$$

where $X$ is $^{13}C$ or $^{15}N$. Excess and content of total organic carbon and nitrogen (TOC and TN per individual) were used to calculate incorporated isotopes ($I_{iso}$) derived from the food source:

$$I_{iso} = E \times TOC \ (or \ TN) \tag{3}$$

The amount of pC ($\mu$g ind$^{-1}$) and pN ($\mu$g ind$^{-1}$) within foraminiferal cytoplasm was calculated as follows (Hunter et al., 2012):

$$pX = \frac{I_{iso}}{\left(\frac{atom\%X_{phyto}}{100}\right)} \tag{4}$$

### 2. 7 Statistical analysis

Experiment 1: The temperature effect on pC and pN within the foraminiferal cytoplasm, and pC:pN was tested using permutation tests and pairwise permutation tests for post-hoc testing (r package rcompanion). Homogeneity of variances was

195 tested using Fligner Killeen test. Relationships of pC and pN after feeding and starvation were explored using linear regression for both species, to observe if pC and pN processing are coupled processes in the two species. Finally, the relative amount of food source-derived carbon and nitrogen after 24 hours starvation was evaluated, to compare the metabolic carbon and nitrogen loss from the two species during the period without food.

Experiment 2: To describe and compare uptake dynamics for the different diets, Michaelis Menten curves were applied on pC

and pN data. The models were tested by applying the lack-of fit method (R package drc). To compare pC and pN values for both diets, pN was calculated from pC for *D. tertiolecta*, and pC from pN for *P. tricornutum*. Hereby acquired estimates for pC and pN might be underestimated or overestimated respectively, due to possible differences in the ratios of carbon:nitrogen excretion or remineralisation, respectively.

### 3 Results

### 3.1 Experiment 1: Nutrient demand and temperature response of *A. tepida* and *H. germanica*

Phytodetrital pC and pN levels derived from *P. tricornutum* detritus was $2-5$ times higher in *A. tepida* compared to *H. germanica* (Fig. 2 a, b). Different incubation temperatures resulted in significant effects on pC levels after 24 hours feeding and 24 hours starvation in both species. *Ammonia tepida* showed a significantly lowered pC content when feeding at 25°C (Fig. 2 a, *A. tepida*, 24 hrs fed, $p < 0.05$). The 24 hour incubation period with no food resulted in significantly lowered pC

levels at 20°C and 25°C (Fig. 2 a, *A. tepida*, 24 hrs starved, $p < 0.05$). In *H. germanica*, the 24 hours feeding period had a similar effect like on *A. tepida*, resulting in significantly lowered pC levels at 25°C (Fig. 2 a, *H. germanica*, 24 hrs fed, $p < 0.05$). A strong effect of increased temperature after the starvation period was present at 25°C (Fig. 2 a, *H. germanica*, 24 hrs starved, $p < 0.05$).

The pN levels in *A. tepida* were considerably affected by temperature after feeding and starvation, whereas there was no

apparent effect on *H. germanica* pN levels, neither after feeding, nor after incubation without food (Fig. 2 b). *Ammonia tepida* reacted with simultaneously lowered pN and pC levels at 25°C after feeding and starvation (Fig. 2 b, *A. tepida*, $p < 0.05$).

The ratios of pC:pN were affected by temperature in both species during feeding and starvation (Fig. 2 c, p < 0.05). Increased temperatures promoted a drop of pC:pN ratios in *A. tepida* during the starvation period (Fig. 2 c, *A. tepida*, p < 0.05). In contrast, temperature specific pC:pN ratios in *H. germanica* showed no change between the incubations with food (24 hrs fed), and the starvation period (24 hrs starved; Fig. 2 c, *H. germanica*). Ratios of C:N show significant temperature related changes in *H. germanica* (p < 0.05), but not in *A. tepida* (Fig. 2 d). The relatively high pN content in *A. tepida* also shows a steeper relationship of cytoplasmic pN and pC, compared to *H. germanica* (Fig. 3 a). Further, there is a far higher metabolic turnover of pC and pN in *A. tepida than in H. germanica*, specifically at 20°C (Fig. 3 b).

**3.2 Experiment 2: Feeding preferences of *A. tepida***

Michaelis Menten curves fitted with no significant deviation of variance within the sample replicates. Enrichment of algal nutrients in foraminiferal cytoplasm were highest when a single diet of *D. tertiolecta* was available (Fig. 4 a). Here, saturation levels (max. 180 ng C ind$^{-1}$) were already reached within three hours after detritus introduction and half saturation with pC in *A. tepida* was reached after 0.6 hours (Table 2). In contrast, a single *P. tricornutum* diet resulted in a slower food intake (Fig. 4 b), with a half saturation of pN levels after 1.4 hours (Table 2). Further, diatom phytodetritus intake resulted in lower levels of pC (max. ~ 80 ng C ind$^{-1}$). In the mixed feeding approach, half saturation of chlorophyte pC was reached after 1.4 hours and diatom pN half saturation was already reached after 0.1 hours. Further, the maximum pC levels of the chlorophyte diet still reached ~ 70 % of those in the single chlorophyte diet, whereas the pN levels of the diatom diet only reached about 30 % of those in the single diatom diet (Fig. 4, Table 2). Chlorophyte intake was faster and higher, both in the single and mixed diet, and diatom pN stagnated already after less than 1 hour in the mixed diet, but after this time period, chlorophyte detritus intake in the mixed diet had continued with increasing pC levels, saturating between 6 and 10 hours (Fig. 4 a, b).

**3.3 Relevance of *A. tepida* and *H. germanica* in intertidal OM fluxes**

Data for the live foraminiferal community in 2016 from the three stained sediment cores showed a typical, low biodiversity mudflat community consisting of *A. tepida*, *H. germanica* and very low abundances of *Elphidium williamsonii* (< 1258 ind m$^{-2}$, all size fractions). Abundances of *A. tepida* and *H. germanica* were equal and decreased with increasing size fraction. The calculated total biomass of live foraminifera in units of TOC is max. ~ 120 mg C m$^{-2}$ (both species, all size fraction, Table 3). From combining in situ abundances and pC values from Experiment 1 (15°C), this foraminiferal community has the potential to take up at least 4 ~ mg C m$^{-2}$ d$^{-1}$, when taking only diatom detritus into account. The contribution of *H. germanica* to this OM processing is only at about 15 %.

**4 Discussion**

Different ecologic lifestyles or adaptations to environmental parameters are important organismic attributes to avoid inter- and intra-specific competition. Further, different metabolic adaptations result in species-specific rates of organic matter turnover.

Our results clearly demonstrate, that food resource partitioning and different temperature adaptations contribute to the fluctuating, temporal distribution and abundance of *A. tepida* and *H. germanica*. Due to these specific adaptations, both species play different roles in intertidal organic matter fluxes. There are, however, limitations for the interpretation of results derived from laboratory incubations. A laboratory setup cannot reproduce natural conditions completely. Therefore, the foraminiferal responses might deviate slightly from their natural behaviour. However, laboratory experiments enable the analysis of the direct response of specimens to a single factor, while maintaining other factors stable. To enable a compatible comparison, we incubated freshly sampled individuals at stable, near natural conditions. Both tested food sources are considered good food sources for intertidal foraminifera (Lee et al., 1966). *Dunaliella tertiolecta* is commonly used in feeding experiments with foraminifera due to its easy culturing. *Phaeodactylum tricornutum*, which represents a more stable (due to the silicate frustule) source of OM, is a common food source of intertidal foraminifera (Murray, 1963). Additional tested food sources would give a more comprehensive picture, but there were limitations in time and material. In the following sections, our results are discussed with respect to these restrictions.

### 4.1 Experiment 1: Nutrient demand and temperature response of *A. tepida* and *H. germanica*

Experiment 1 shows clear differences in the amount of phytodetritus intake and different carbon and nitrogen budgeting between the two species (Fig. 2, Fig. 3). *Ammonia tepida* has a higher affinity to the diatom detritus food source with a three times higher intake of diatoms at the two lower temperatures compared to *H. germanica*. This lower food intake by *H. germanica* could be explained by the mixotrophic lifestyle of this species. *Haynesina germanica* is known to host kleptoplasts, exploiting the photosynthetic activity of ingested chloroplasts as an additional energy source (Lopez, 1979; Pillet et al., 2011). This species might therefore utilize nutrients (carbohydrates) derived from the photosynthetic activity of incorporated chloroplasts (Cesbron et al., 2017). This lifestyle could cause a lower demand for and lower turnover of OM as food source (Cesbron et al., 2017). In our study, the pC intake in *H. germanica* was ~ 67% lower than that of *A. tepida* (Fig. 2). Highly specialized sea slugs use plastids as energy reservoirs at times of low food availability (Cartaxana et al., 2017; Hinde and Smith, 1972; Marín and Ros, 1993), where carbon supply from chloroplasts can cover 60% of total carbon input (Raven et al., 2001). In kleptoplast hosting sea slugs, free $NH_4^+$ from the seawater is a primary source for the generation of amino acids via kleptoplast metabolism within the slug (Teugels et al., 2008). A similar mechanism in *H. germanica* might explain the high relative turnover of pN (Fig. 3b). Phytodetrital nitrogen might therefore be disposed at a higher rate in a relatively temperature independent process, probably in the form of dissolved organic nitrogen, further causing higher pC:pN ratio in the cytoplasm of *H. germanica* (Fig. 2).

In addition to the higher rates of phytodetritus intake, *A. tepida* shows a considerably higher metabolic turnover of pC and pN than *H. germanica* (Fig. 3b). According to Cesbron et al. (2016), respiration rates (normalized to pmol mm$^{-3}$ d$^{-1}$) are about 2 – 12 times higher in *A. tepida* specimens than in *H. germanica* specimens from the same location. In this study, a 4 - 7 times higher release of phytodetritus-derived pCper individual and day (size fraction 250 – 355 µm) was observed in *A. tepida*. Interestingly, this study shows similar reactions of both species in carbon loss due to increased temperature. An earlier study

on the temperature effect on *D. tertiolecta* detritus intake of the two species showed a higher sensitivity to increased temperatures in *H. germanica*, and far lower rates of chlorophyte detritus intake compared to this study (Wukovits et al., 2017). In contrast, *A. tepida* seems to be more tolerant to higher temperatures when feeding on chlorophyte detritus. The results of Experiment 1 suggest a niche separation of the two species with respect to phytodetritus or OM availability and temperature.

## 4.2 Experiment 2: Feeding preferences of *A. tepida*

The findings of Experiment 2 suggest that *A. tepida* might prefers OM food sources, which are easy to exploit and to break down. The high intake values in the *D. tertiolecta* mono-diet one hour after incubation and the saturation of cytoplasmic pC levels after three hours indicate a high affinity to chlorophyte detritus (Fig. 4, Table 2). Earlier studies also observed quick and high ingestion rates of chlorophyte detritus (*Chlorella* sp.) by the genus *Ammonia* (Linshy et al., 2014; Wukovits et al., 2017, 2018). The fast saturation with diatom detritus after one hour in the mixed diet and the advanced and high intake of *D.*
*tertiolecta* could even indicate an avoidance of *P. tricornutum* and selective feeding on *D. tertiolecta*. Probably, the soft cells of chlorophytes enable a faster and easier metabolic processing of this food source compared to the harder diatom frustules. The recognition of such food sources could be achieved by chemosensory behaviour of the foraminifera (cf. Langer and Gehring, 1993) and the attraction to specific substances attached to, or leaking from the food particles, similar to some other protists, which react to food-specific amino acids (Almagor et al., 1981; Levandowsky et al., 1984). Microalgal communities
in tidal sediments typically consist of microphytobenthic diatoms, which are considered to be the main food source for intertidal foraminifera. An isotope labelling study has shown that diatoms (*Navicula salinicola*) are taken up by *A. tepida* at high rates, but the complete release of the content of the diatom frustules can take several days (LeKieffre et al., 2017). This might not fit the nutrient demands of *A. tepida* at times of high metabolic activity. Therefore, a shift from microphytobenthos to particulate OM from riverine or tidal transport might be a feeding strategy in *A. tepida*. Specifically at higher temperatures, when more
energy is needed to maintain metabolic activities.

     In general, food sources of *A. tepida* include microalgae, phytodetritus, bacteria and sometimes metazoans (Bradshaw, 1961; Dupuy et al., 2010; Moodley et al., 2000; Pascal et al., 2008). Bacteria are considered to play a minor role in the diet of *A. tepida* (Pascal et al., 2008), and reports on metazoan feeding in *A. tepida* are restricted to a single observation (Dupuy et al., 2010). In contrast to *A. tepida*, *H. germanica* does actively ingest bacteria and they can occasionally be preferred over diatoms
(Brouwer et al., 2016). Diatoms are reportedly taken up by *H. germanica*, and conical test structures serve as tools to crack diatom frustules open (Austin et al., 2005; Ward et al., 2003). These chloroplasts derived from diatoms remain as functional kleptoplasts, as mentioned above, within the cytoplasm of *H. germanica*.

## 4.3 Relevance of *A. tepida* and *H. germanica* for intertidal OM fluxes

     Data of foraminiferal abundances or foraminiferal biomass are important variables to estimate foraminiferal nutrient fluxes. In
this section, we discuss the relevance of *A. tepida* or *H. germanica* in intertidal fluxes of phytodetrital carbon and nitrogen as estimated from sediment core data in combination with results from the laboratory feeding experiments of this study. The total

biomass of the two species in the sampling area ranges between ~ 116 and > 380 mg TOC m$^{-2}$ (size fraction 125-355 μm) at the sampling dates in late April/early May in two consecutive years (Table 3). This lies within the range of estimations for hard-shelled foraminifera in other areas of the Wadden Sea (van Oevelen et al., 2006b, 2006a, TOC max. ~ 160 - 750 mg C

m$^{-2}$). Our phytodetritus uptake estimates propose, that the foraminiferal biomass consists of ~ 6 – 8% diatom-derived pC /TOC, with the major amount contained within *A. tepida* (compare Table 3). An *in-situ* feeding experiment with deep-sea foraminifera resulted in values of ~ 1 – 12% pC/TOC (Nomaki et al., 2005b). Similar *in-situ* incubations in the core of the oxygen minimum zone of the Arabian Sea report ~ 15% pC/TOC in epifaunal and shallow infaunal foraminiferal carbon uptake (Enge et al., 2014). *In-situ* incubations offer results closest to the natural responses of organisms in their natural habitat and enable precise

estimates of foraminiferal nutrient fluxes. Although, specific microhabitat conditions can have a strong influence on organismic behaviour. The artificial conditions in laboratory experiments also have an influence on physiological analysis, therefore the obtained results should be treated with caution. However, our estimates lie in the same order of magnitude as the above mentioned *in-situ* studies and offer a basis for estimations on foraminiferal carbon and nitrogen fluxes. General variations in foraminiferal carbon and nitrogen budgets can be caused by different adaptations to variable food availability in

different habitats. This can be achieved by different controls of energy metabolism (e.g. Linke, 1992) or different trophic strategies (e.g. Lopez, 1979; Nomaki et al., 2011; Pascal et al., 2008). Our results suggest, *A. tepida* has a higher relevance for intertidal OM processing than *H. germanica*. This can be mainly attributed to the sequestered chloroplasts within the cytoplasm of *H. germanica*. Kleptoplasty is a wide spread phenomenon in foraminifera, specifically in species inhabiting dysoxic sediments, where kleptoplasts could promote survival in anoxic pore waters (Bernhard and Bowser, 1999). They might be

involved in biochemical pathways within the foraminiferal cytoplasm, e.g. the transport of inorganic carbon and nitrogen (LeKieffre et al., 2018). Further, transmission electron microscopic investigations on *H. germanica* report a very limited abundance of food vesicles (Goldstein and Richardson, 2018). Kleptoplast-bearing species might occupy a distinct niche concerning their energetic demands. Additionally, they might play a not yet discovered importance in the fluxes of inorganic or dissolved carbon and nitrogen compounds. However, secondary producers with high uptake rates and a quick response to

particulate OM sources like *A. tepida* play a strong role in the biogeochemical carbon and nitrogen recycling.

The high rates of OM carbon and nitrogen turnover are mainly caused by *A. tepida* populations (Table 3). The process of carbon and nitrogen regeneration by OM remineralisation plays an important role in marine biogeochemical cycling. Carbon loss, e.g. due to organismic respiration or OM remineralisation to $CO_2$, reduces the availability of organic carbon sources in the heterotrophic food web. As mentioned above, in the heterotrophic, coastal zone 30% of the carbon pool are lost as via

respiration. Whereas, dissolved organic carbon sources from organismic excretion can serve as an important nutrient source for bacteria (Kahler et al., 1997; Snyder and Hoch, 1996; Zweifel et al., 1993). Therefore, the fast processing of OM in *A. tepida* might be an important sink for inorganic carbon ($CO_2$ respiration) and at the same time a link for dissolved organic carbon sources in intertidal carbon and nitrogen fluxes. According to this study, maximum pC flux through *A. tepida* can reach values of ~ 36 mg C m$^{-2}$ d$^{-1}$ when feeding on chlorophytes at 20°C (estimated from Experiment 2, Fig. 3 relative release, and

max. abundances). Therefore, *A. tepida* could contribute up to 10% of the turnover of OM derived from gross particulate

phytoplankton production at the sampling date in April/May 2016, with a gross particulate primary production between ~ 230 – 1500 mg C m$^{-2}$ d$^{-1}$ (Tillmann et al., 2000). This is comparable with the study of Moodley et al., (2000), where *Ammonia* sp. incorporated ~ 7% within 53 hours in sediment core incubations feeding experiments in sediment incubations with added, labeled chlorophyte detritus.

Planktonic protozoa are the primary regenerators of marine nitrogen, transforming OM-nitrogen to their primary N-excretion product, $NH_4^+$ (Glibert, 1997). The excretion of $NH_4^+$ by marine protists can contribute to a large part to the nutritional demands of marine primary productivity (Ferrierpages and Rassoulzadegan, 1994; Ota and Taniguchi, 2003; Verity, 1985). Nitrogen regeneration by protozoa was supposed to play a far higher role than bacterial nitrogen regeneration in the marine microbial food chain (Goldman and Caron, 1985). Indeed, excreted nitrogen can serve as important nutrient sources for microbes

(Wheeler and Kirchman, 1986). The release of dissolved organic nitrogen and $NH_4^+$ by e.g. copepods, can be a major driver for marine microbial production (Valdés et al., 2018). Here, foraminiferal nitrogen excretion values are in the range of estimations for weight-specific $NH_4^+$ excretion in marine protozoa according to Dolan (1997) (data for foraminiferal weight, comp. supplementary Fig. 2). Due to their high abundances, nitrogen release by *A. tepida* as observed in this study could reach 2.5 mg N m$^{-2}$ d$^{-1}$ or ~ 73 nmol N dm$^{-2}$ h$^{-1}$, respectively, at 15°C and high diatom availability (comp. Table 3). As a rough

estimate for *A. tepida* feeding at high abundances and high availability of chlorophyte detritus at 20°C, these values could increase to ~ 22 mg N m$^{-2}$ d$^{-1}$ or ~ 0.6 µmol N dm$^{-2}$ h$^{-1}$ (Fig.1, Table 3). Therefore, foraminiferal nitrogen release as $NH_4^+$ or amino-acids could cover a considerable amount of the nutritional nitrogen demand in marine bacteria (cf. Wheeler and Kirchman, 1986), which assimilate $NH_4^+$ (and amino acid-derived $NH_4^+$) to sustain their glutamate-glutamine cycle. Vice versa, the labile dissolved organic matter derived from bacterial decomposition of refractory organic matter provides a valuable food

source for some benthic foraminifera, and is indispensable for the reproduction of some foraminiferal species (Jorissen et al., 1998; Muller and Lee, 1969; Nomaki et al., 2011). In many marine diatoms, which are the main drivers of marine primary productivity, $NH_4^+$ is the preferred source for nitrogen uptake over $NO_3^-$ (Sivasubramanian and Rao, 1988). Foraminifera could act as important nutrient providers for closely associated diatoms, which are also considered as one of their main food sources (Lee et al., 1966). Consequently, the kleptoplast-hosting metabolism in *H. germanica* could benefit from regenerated nitrogen

sources by the high OM mineralization rates in *A. tepida*. In summary, foraminiferal carbon and nitrogen fluxes constitute an important link in the food web complex of primary consumers and decomposers.

## 5 Conclusions

This study compares differences in the feeding behaviour, nutrient demand, and OM flux of two intertidal foraminiferal species. Our results clearly show that *A. tepida* has a higher impact on the fluxes of phytodetrital carbon and nitrogen in

intertidal sediments than *H. germanica*. This can partly be explained by their different lifestyles. Differences in temperature acclimatization or preferences to different food sources can serve as strategies to avoid spatial and temporal interspecific competition, resulting in a niche separation of the two species with respect to phytodetritus or OM availability and temperature.

Accordingly, *H. germanica* could be associated with environmental conditions of moderate availability of microphytobenthos and lower temperatures, as given prior to the diatom spring bloom. Whereas *A. tepida* could take advantage of seasons characterized by higher input of allochthonous OM. Further, temperature fluctuations in combination with allochthonous OM availability have less effect on the carbon and nitrogen processing in *A. tepida*. These differentiations in their metabolic OM processing and lifestyles suggest a far higher relevance of *A. tepida* in the mediation of the fluxes of intertidal carbon and nitrogen.

Acknowledgements: We thank Patrick Bukenberger and Murtaza Kulaksiz for help with sampling and culture maintenance; and Margarete Watzka for EA-IRMS analysis. We also thank the two anonymous referees for their contribution to the improvement of the early version of the manuscript. Open access funding provided by University of Vienna.

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

**Table 1. Experimental setup and conditions**

| | species | Individuals per replicate | Sampling intervals [h] | T [°C] | food source | amount of food added [mg C m$^{-2}$] | amount of food added [mg N m$^{-2}$] |
|---|---|---|---|---|---|---|---|
| Exp. 1 | *A. tepida* | 50 - 55 | 24 / fed 24 / starved | 15, 20, 25 | Diatom | 540 | 100 |
| | *H. germanica* | 50 - 55 | 24 / fed 24 / starved | 15, 20, 25 | Diatom | 540 | 100 |
| Exp. 2 | *A. tepida* | 55 | 1, 3, 6, 12, 24 | 20 | Chlorophyte | 410 | 71 |
| | *A. tepida* | 55 | 1, 3, 6, 12, 24 | 20 | Diatom | 647 | 21 |
| | *A. tepida* | 55 | 1, 3, 6, 12, 24 | 20 | Chlorophyte + Diatom | 206 + 324 | 35 + 10 |

**Table 2. Michaelis Menten parameters of curves for pC and pN intake in Figure 4 (bold font = data from measured values, regular font = data from calculated values, Vmax = maximum pC/pN; Km = half saturation for pC/pN, Res. SE = residual standard error, DF = degrees of freedom).**

| | | Vmax | Km | Res. SE | DF |
|---|---|---|---|---|---|
| pC | **Chlorophyte mono diet** | **179.875** | **0.611** | **20.745** | **16** |
| | **Chlorophyte mixed diet** | **124.196** | **1.359** | **11.918** | **15** |
| | Diatom mono diet | 80.191 | 1.374 | 9.290 | 16 |
| | Diatom mixed diet | 24.000 | 0.098 | 2.983 | 16 |
| | | | | | |
| pN | Chlorophyte mono diet | 30.860 | 0.611 | 3.559 | 16 |
| | Chlorophyte mixed diet | 21.307 | 1.359 | 2.286 | 12 |
| | **Diatom mono diet** | **10.912** | **1.374** | **1.264** | **12** |
| | **Diatom mixed diet** | **3.267** | **0.100** | **0.410** | **16** |

**Table 3. Mean abundances (± SD)of live *A. tepida* and *H. germanica* (0-1 cm sediment depth), TOC, TN, and carbon and nitrogen flux calculated from sediment cores (early May 2015* n = 1, late April 2016 n = 3). Data for 15°C of Experiment 1 were used to estimate carbon and nitrogen fluxes (n.d. = not determinded).**

| | size fraction [µm] | abundance [ind m$^{-2}$] | TOC [mg m$^{-2}$] | TN [mg m$^{-2}$] | pC$_{intake}$ [mg C m$^{-2}$ d$^{-1}$] | pC$_{release}$ [mg C m$^{-2}$ d$^{-1}$] | pN$_{intake}$ [mg N m$^{-2}$ d$^{-1}$] | pN$_{release}$ [mg N m$^{-2}$ d$^{-1}$] |
|---|---|---|---|---|---|---|---|---|
| *A. tepida* 2015 [1] | 125 - 250 | 1166979 | 226.516 | 77.322 | 20.937 | 8.375 | 5.333 | 1.813 |
| | 250 - 355 | 186742 | 163.428 | 35.817 | 11.467 | 4.480 | 1.919 | 0.651 |

| | | | | | | | | |
|---|---|---|---|---|---|---|---|---|
| | >355 | 3773 | n.d. | n.d. | n.d. | n.d. | n.d. | n.d. |
| *A. tepida* | 125 - 250 | 97248 (±10471) | 21.317 | 7.277 | 1.745 | 0.698 | 0.444 | 0.151 |
| 2016 | 250 - 355 | 43594 (±11041) | 38.152 | 8.361 | 1.802 | 0.704 | 0.302 | 0.102 |
| | >355 | 4401 (±12786) | n.d. | n.d. | n.d. | n.d. | n.d. | n.d. |
| *H. germanica* | 125 - 250 | 109823 (±54078) | 25.717 | 6.867 | n.d. | n.d. | n.d. | n.d. |
| 2016 | 250 - 355 | 29342 (±12768) | 30.978 | 5.311 | 0.601 | 0.188 | 0.069 | 0.028 |
| | >355 | 3773 (±2741) | n.d. | n.d. | n.d. | n.d. | n.d. | n.d. |

[1] Data from Wukovits et al. (2018)

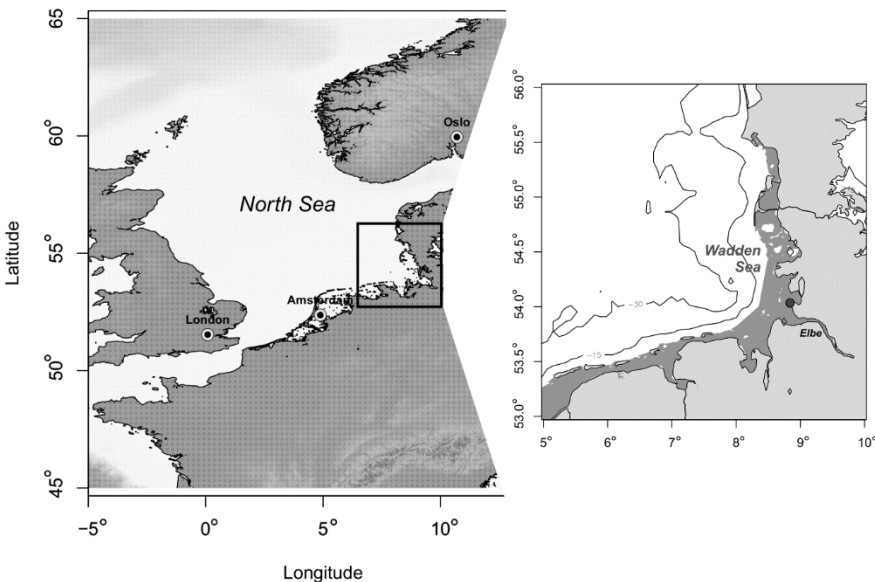

**Figure 1. Sampling area.**

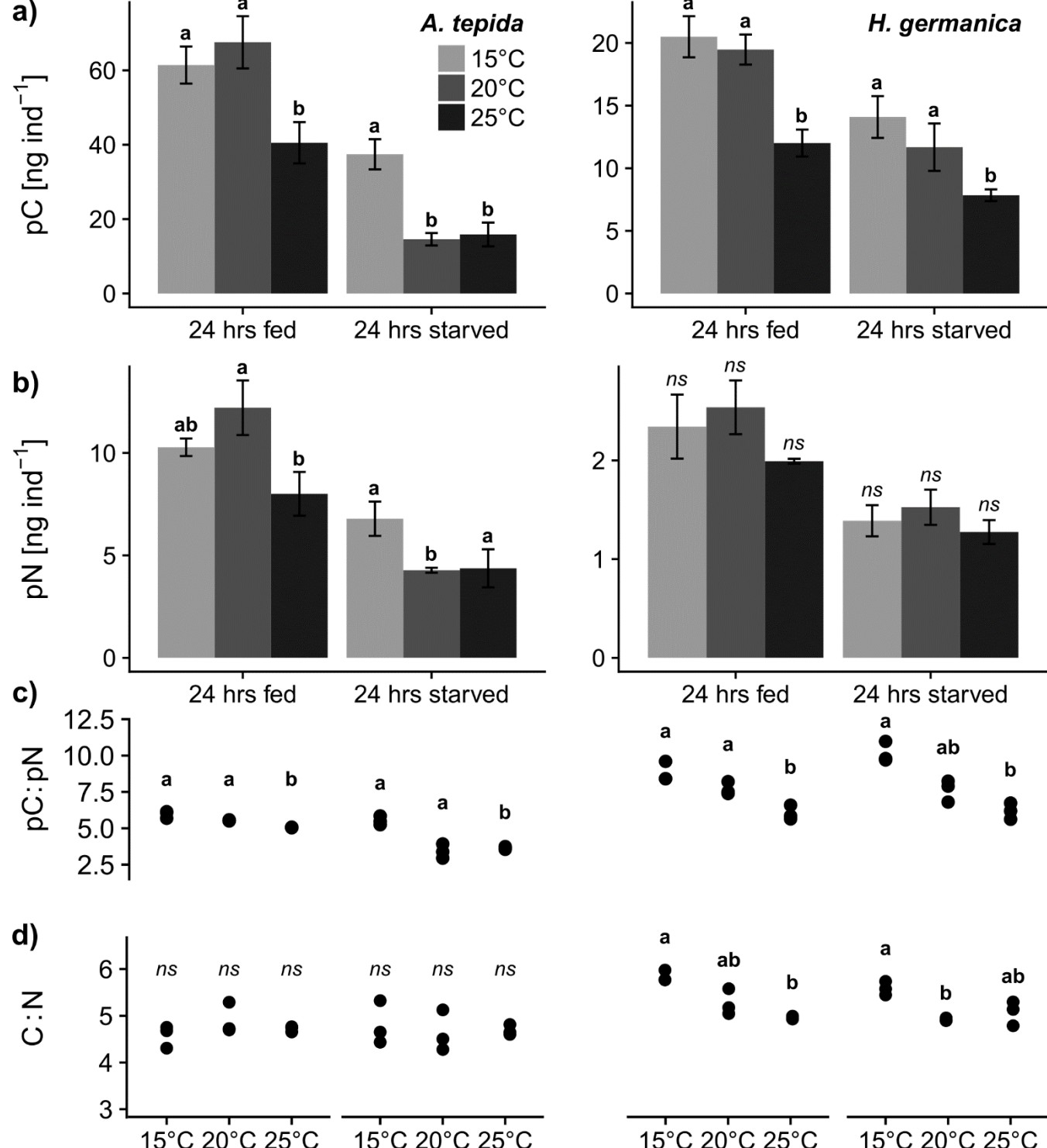

Figure 1 a – d). Comparison of pC and pN from diatom feeding in *A. tepida* and *H. germanica* after a 24 hours feeding period (24 hrs fed) and 24 hours without food (24 hrs starved) at 15°C, 20°C, and 25°C. Letters show significant differences of a) cytoplasmic

pC; b) pN between incubation temperatures within the 24 hours feeding period/24 hrs fed and the 24 hours incubation without food/24 hrs starved; c) pC : pN ratio (n=3, in all cases); d) ratios of foraminiferal cytoplasmic C:N ratios; p < 0.05, pairwise permutation tests, *ns* = not significant

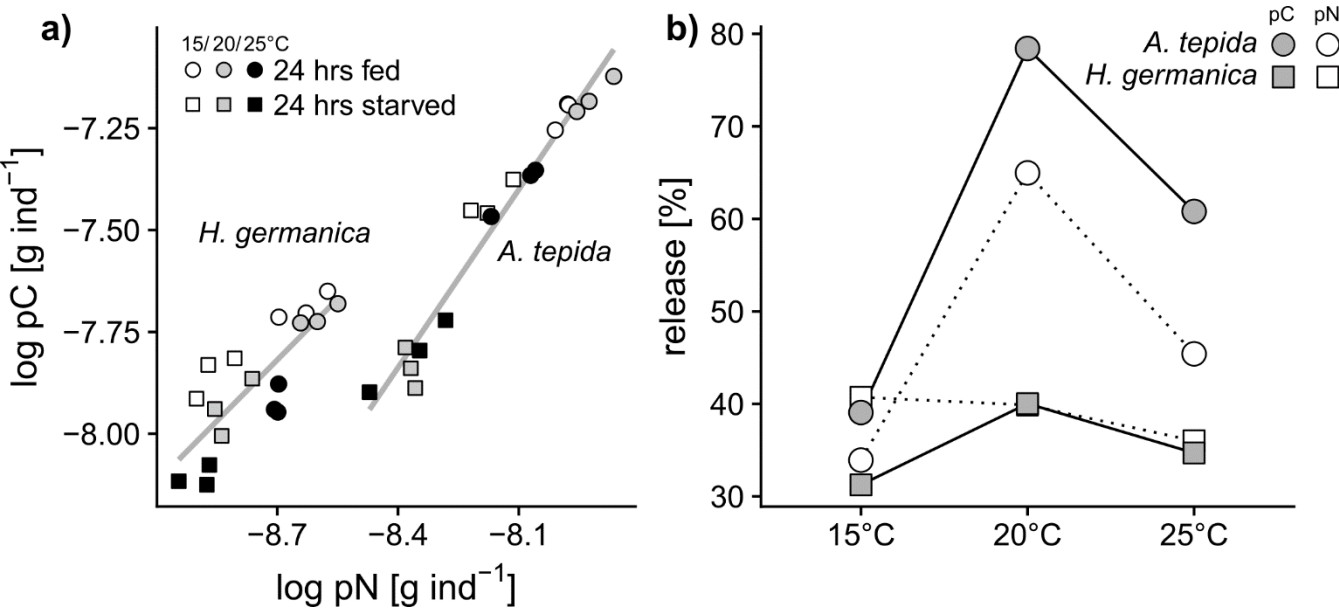

Figure 2 a-b. a) Relationship of pC and pN in *A. tepida* and *H. germanica* (*A. tepida*: $R^2 = 0.96$, y = 1.5x + 4.4, $p < 0.01$; *H. germanica*: $R^2 = 0.64$, y = x + 0.88, $p = 0.011$), and b) phytodetrital carbon and nitrogen turnover as percent release (of total intake of pC or pN per day, respectively).

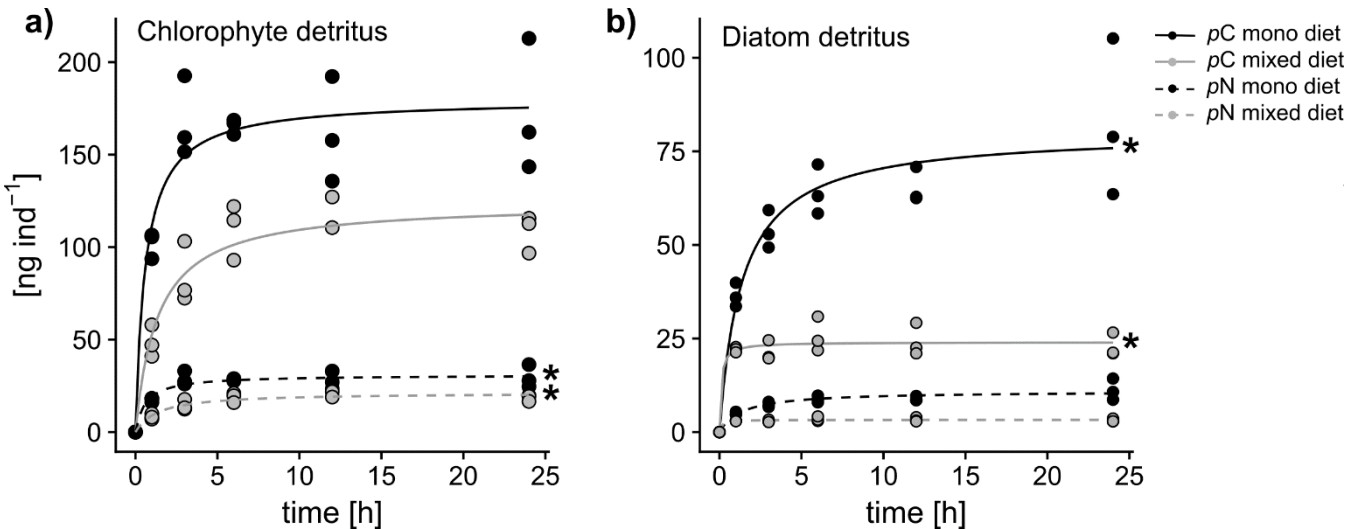

Figure 3 a-b. Comparison of chlorophyte and diatom phytodetritus feeding in *A. tepida* for 24 hours, presenting feeding dynamics for a) chlorophyte detritus and b) diatom detritus. Curves show Michaelis Menten Fits through triplicates for each approach (stars indicate calculated values for pC or pN).