# Peer review of "The distinct roles of two intertidal foraminiferal species in phytodetrital carbon and nitrogen fluxes - results from laboratory feeding experiments"

_Biogeosciences, 2018_

## Referee Comment (RC1) · Anonymous Referee #1 · 4 Jun 2018

In this paper, Wukovits et al. report the results of two series of short-term experiments involving Ammonia tepida and Haynesina germanica, two benthic foraminiferal species commonly found in intertidal environments. In particular, three different temperatures (15°C, 20°C, and 25°C) and two food sources (diatom vs. chlorophyte) were tested in this study. The goal of these experiments was to investigate carbon and nitrogen uptake and turnover by A. tepida and H. germanica and the impact of these processes on intertidal organic matter fluxes. Overall, the manuscript is well written and the findings are interesting, in particular when discussed in the context of the carbon and nitrogen

cycles in intertidal environments. However, I do have some issues with this work, as I explain more in detail below. Because of this, I recommend major revisions before the manuscript can be accepted for publication in "Biogeosciences".

Major comments

Introduction. I think that the manuscript would benefit from a greater overview of: 1) the carbon and nitrogen cycles in coastal environments 2) the role of benthic foraminifera in the carbon and nitrogen cycles Some information are provided in the Discussion section of the manuscript. However, I think that a general overview of these processes should be included in the Introduction, as well.

Line 49. The authors briefly mention previous studies on feeding preferences/strategy. Considering that these are important points that are discussed later in the manuscript, I suggest providing more information regarding past experimental studies. In doing so, the authors can better emphasize the novelty of their work in the context of earlier investigations.

Line 56. This work might be of interest to readers who might not be familiar with foraminifera. Thus, I recommend to better explaining what the authors mean by "release of OM derived carbon and nitrogen in foraminifera" and how this connects with OM remineralization processes in coastal environments.

Material and Methods. I think that the authors should provide more information regarding their experimental design. For example, for Experiment 1, why did they choose to terminate the incubation after 24 hours? Is this enough time to obtain a significant result? Why chlorophyte as a food source was not tested in Experiment 1? Why H. germanica was not included in Experiment 2? Why was 20°C (and not 15°C or 25°C) the temperature tested in Experiment 2?

Lines 88-90. How were these atom%'s established?

Line 93. Is 28 PSU the same salinity as at the sampling site?

Lines 103-109 and 124-128. My suggestion is to explain the statistical treatment of the data in a separate section.

Line 132. "The sediment core data, together with the data from the laboratory experiments, were used to estimate (. . .)" The authors combined sediment core data with data from laboratory experiments to estimate total foraminiferal biomass and foraminiferal C and N processing. My question is why? The data obtained from sediment core ("natural samples") should be compared (and not combined) with the ones obtained from the laboratory experiments, as experiments are a simplification of the natural environment.

Line 140. After decalcification, the authors kept the foraminiferal at 50°C to dry for three days. Are the authors using a published protocol? If so, please cite the reference. If not, is it possible that such a long drying step could have altered their results?

Table 1, 2nd column. "50 – 55". Are 50 the number of specimens used in the 24/fed experiment and 55 the number of specimens used in the 24/starved experiment? If so, please specify. [h] should be [hrs] for consistency with the rest of the manuscript.

Results. I invite the authors to consider reporting the data presented in figure 1 as an additional (supplementary?) table.

Figure 1 c and d. Considering that the temperature is specified in the x axis, I do not think that the authors need to color code the data points, also because the "middle" shade of gray and the darker shade of grey cannot be easily distinguished. An alternative might be using different symbols for different temperatures. Also the meaning of "ns" is not included in the caption.

Figure 2a. Can the data be differentiated based on the temperature of the experiments? Maybe different symbols (or colors) can be used for this purpose.

Figure 2b. The figure is a bit confusing. Again, I would recommend using different symbols (or colors) for different trends.

Figure 3. Chlorphyte should be Chlorophyte. Also, not all the symbols of the figure

legend correspond to the symbols on the plots.

Section 3.2 I am not sure I understood how the authors were able to distinguish the contribution of one food source over the other in the mixed diet.

Discussion. The authors mention the presence of chloroplasts in Haynesina germanica. How about Ammonia tepida (cf. Jauffrais et al., 2016)?

Lines 294-296. I think that the authors make a very interesting point here. Can they expand on this?

Minor comments

Line 12. Should "13C & 15N" be "13C and 15N"? This comment applies to the rest of the manuscript.

Lines 14-19. Throughout the manuscript, the results obtained in A. tepida are discussed before those obtained in H. germanica. I recommend maintaining the same structure in the abstract, as well.

Line 25. "Coastal sediments represent the largest pool of marine particulate organic matter (OM) (...)". Can the authors add some numbers (maybe a percentage?) regarding how big the OM pool is in coastal sediments? In my opinion, such a number will provide a good context to discuss the data obtained from the experiments and to discuss the importance of remineralization processes mediated by benthic foraminifera in coastal environments.

Line 36. "(...) e.g., temperature or OM quality". This should be "temperature and/or OM quality".

Lines 40-41 and 47-48. These sentences are not very clear. Please, rephrase.

Lines 58-59. Considering that the experiment described at lines 58-59 is Experiment #2, I suggest moving this sentence after the sentence at lines 60-61, which refers to Experiment #1.

Line 72. M2 should be m2.

Line 75. "Individuals were picked from the sediment in sufficient and collected (. . .)". In sufficient number?

Line 77. "Dunaliella tertiolecta and Phaeodactylum tricornutum" The scientific name was already defined at line 58, so this should be D. tertiolecta and P. tricornutum. This comment applies to the rest of the manuscript, with the exception of tables and figures.

Lines 79-80. "The experiments started after accumulation of sufficient foraminiferal material three weeks after the field sampling." I assume the authors achieved foraminiferal reproduction during the initial incubation. If my assumption is correct, then it would be good to specify so and provide some information about the conditions used to maintain the foraminifera prior the beginning of the experiments. If the authors know, it might be of interest to know how successful the reproduction event was.

Line 84. NaH13CO3, Na15NO3 should be NaH13CO3, Na15NO3.

Line 88. C.f. should be cf. This comment applies to the rest of the manuscript.

Line 108. What do the authors mean with "carbon and nitrogen costs of the two species during the period without food"?

Line 114. Cm-2 should be cm-2.

Line 135. A parenthesis is missing.

Line 137. I suggest including the word "cytoplasm" prior "isotope analysis", for clarity.

Line 153 (formula #2). atomXsample – should this be atom%Xsample? Same for background.

Line 155. I recommend writing the Iiso formula as the other formulas, for clarity.

Line 184. There is an extra period after Table 2.

Line 205. No comma needed.

Line 212. Phaedactylum tricornutum shoud be in italic.

Section 4.1 revise references – e.g., a comma is missing between the authors' names and the year of publication and a semicolon should be used to separate different references.

Line 232. Missing parenthesis.

Line 250. Almagor et al. – publication year 1981.

Lines 281 and 290. Missing parentheses around the year of publication.

Lines 288 and 295. Comp. should probably be cf.

Reference. Jauffrais, T., Jesus, B., Metzger, E., Mouget, J.-L., Jorissen, F., and Geslin, E.: Effect of light on photosynthetic efficiency of sequestered chloroplasts in intertidal benthic foraminifera (Haynesina germanica and Ammonia tepida), Biogeosciences, 13, 2715-2726, https://doi.org/10.5194/bg-13-2715-2016, 2016.

––––––––––––––––––––––––––––––

---

## Referee Comment (RC2) · Anonymous Referee #2 · 24 Aug 2018

General Summary: This is a valuable contribution to our understanding of Foraminifera and their role in benthic carbon and nitrogen turn-over in the intertidal environment. The study is based on laboratory experiments, focusing on two species of intertidal benthic Foraminifera, with potential to inform and improve our understanding of their role in carbon and nitrogen cycling. One of these species (A. tepida) may be particularly prone to taxonomic confusion with closely related species (i.e. cryptic diversity) and both species are likely to adopt different feeding strategies, including kleptoplastidy (e.g. Austin et al. 2005; Jauffrais et al. 2016). As such, some challenges remain in

the translation of these experimental data to the real world (consider aspects of niche partitioning/differentiation and their significance in net annual fluxes), but the authors are to be commended on their experimental design in providing an improved understanding of the key processes of carbon/nitrogen intake/uptake. In my view, the work is original, shows innovation and makes a useful contribution to the field of study of benthic Foraminifera and their role within intertidal biogeochemical cycling.

Recommendation: I would recommend acceptance of the manuscript subject to some moderate revision. While the experiments are clearly described, I felt that the rationale to translate these experiments to field-based interpretations were rather limited – I suggest the authors strengthen this aspect of the manuscript, making it clear what the findings mean in terms of field-context by reference to a wider literature. If this is not possible, then the translation of these results from laboratory to field should be treated with greater caution e.g. tone-down statements such as those on line 311. Sections of the manuscript, such as 3.3, are very interesting but take a very linear approach – again, cross-reference to any extended literature might strengthen these arguments. The discussion leaves the reader with a sense of some "loose ends", so again – perhaps some editing of the discussion to focus on a stronger connection between experiments and field would be helpful. Try to avoid, as in the conclusion (section 5), open-ended discussion where the role for bacteria, for example, are never quite tied-down.

Specific Comments: there are quite a few minor grammatical errors; I would recommend a careful proof reading of any resubmitted material. Please ensure that you include a proper and complete review of the recent literature (e.g. Jauffrais et al. 2016) on kleptoplastidy – you can largely include this in the introduction/state-of-the-art; why not take the opportunity to highlight that "uptake" remains a critical feeding strategy and that despite these exciting new developments, the focus of your manuscript illustrates the critical role of benthic Foraminiferal feeding as a key component in the benthic biogeochemical cycle of the intertidal environment – can you say this? Personally, I think

you could develop the illustrations/figures – these can be helpful to the readership and I would be tempted to add more, including a location map and some supplementary SEM images of the species – as noted above the genus Ammonia is particularly problematic and displays cryptic diversity, does it not?

―――――――――――――――――――――

---

## Author Comment (AC1) · 15 Sep 2018

Response to Referee 1:

Major comments

Introduction R1: Introduction. I think that the manuscript would benefit from a greater overview of: 1) the carbon and nitrogen cycles in coastal environments 2) the role of benthic foraminifera in the carbon and nitrogen cycles. Some information are provided in the Discussion section of the Manuscript. However, I think that a general overview

of these porcesses should be included in the Introduction, as well.

JW: The introduction was extended, providing the following information about coastal carbon and nitrogen cycles and the role of foraminifera in these cycles:

Line 26-43: "Oceanic and terrestrial systems are connected by the carbon cycling in coastal waters, which contribute to a major part of the global carbon cycles and budgets (Bauer et al., 2013; Cai, 2011; Cole et al., 2007; Regnier et al., 2013). Estuaries are an important source for organic matter in coastal systems and were estimated to account for $\sim$ 40% of oceanic phytoplankton primary productivity (Smith and Hollibaugh 1993). Most estuarine areas are considered to be net heterotrophic, or act as carbon sinks, respectively (e.g. Caffrey, 2003, 2004; Cai, 2011; Herrmann et al., 2015). In general, 30% of overall coastal carbon is lost by metabolic oxidation (Smith and Hollibaugh 1993). Foraminifera are highly abundant in estuarine sediments and contribute strongly to these processes (Alve and Murray, 1994; Cesbron et al., 2016; Moodley et al., 2000; Murray and Alve, 2000). They feed on various sources of labile particulate OM, including microalgae and detritus, and provide a pivotal link in marine carbon cycles and food webs (Bradshaw, 1961; Goldstein and Corliss, 1994; Heinz et al., 2001; Lee et al., 1966; Lee and Muller, 1973; Nomaki et al., 2005b, 2006, 2009, 2011). The nitrogen compounds of OM particles are usually remineralized to ammonium ($NH_4^+$). In this way, nitrogen gets again available as nutrient for primary productivity. A major part of this process is attributed to prokaryotic degraders, but protists are also involved in the process of regeneration of organic nitrogen compounds (Ferrier‐Pages and Rassoulzadegan, 1994; Ota and Taniguchi, 2003; Verity et al., 1992). Due to their high abundances, we consider, that foraminifera contribute a large part to this OM reworking and the regeneration of carbon and nitrogen compounds from particulate OM sources, e.g. phytodetritus. In this study, we quantify the bulk OM-derived carbon and nitrogen release, which originates rather via excretion of organic carbon and nitrogen compounds (vesicular transport of metabolic waste products), respiration or diffusion of inorganic carbon and nitrogen by these single celled microorganisms."

[Figure]

R1: Line 49: The authors briefly mention previous studies on feeding preferences/strategy. Considering that these are important points that are discussed later in the manuscript, I suggest providing more information regarding past experimental studies. In doing so, the authors can better emphasize the novelty of their work in the context of earlier investigations.

JW: Added section:

Line 70-79: "Laboratory feeding experiments have shown, that A. tepida responds to several food sources, including different live microalgae (chlorophytes and diatoms) and chlorophyte and diatom detritus (Bradshaw, 1961; Lee et al., 1966; LeKieffre et al., 2017; Linshy et al., 2014; Pascal et al., 2008; Wukovits et al., 2017, 2018). On the other hand, H. germanica shows a low affinity to chloroplast detritus food sources (Wukovits et al., 2017), but feeds actively on diatoms (Ward et al., 2003) and takes up inorganic, dissolved C & N compounds (LeKieffre et al., 2018). Both species are found in muddy coastal sediments containing high loads of nutrients or OM (Armynot du Chatelet et al., 2009; Armynot du Châtelet et al., 2004). But considering their different feeding strategies they might play distinct roles in the reworking of OM. Recent literature still lacks direct, quantitative comparisons of foraminiferal species-specific quantitative OM-derived C & N ingestion and release. Therefore, this study aims to compare and quantify variations in their respective uptake of OM (phytodetritus).

R1: Line 56. This work might be of interest to readers who might not be familiar with foraminifera. Thus, I recommend to better explaining what the authors mean by "release of OM derived carbon and nitrogen in foraminifera" and how this connects with OM remineralization processes in coastal waters.

JW: The following sentence was added: Line 41-43: "In this study, we quantify the bulk OM-derived C & N release, which originates rather via excretion of organic carbon and nitrogen compounds (vesicular transport of metabolic waste products), respiration or diffusion of inorganic C and N by the single celled micro-organisms."

Additional changes in the introduction to better integrate the reviewers suggestions:

The following section was removed to keep the introduction concise: "Certain key species in foraminiferal communities contribute with a major extant to the OM processing in extensive, highly productive marine environments (Enge et al., 2014, 2016, Moodley et al., 2000, 2002, Nomaki et al., 2005a, 2008; Witte et al., 2003; Wukovits et al., 2018). Therefore, the quantification of foraminiferal carbon and nitrogen processing derived from OM and food selectivity in foraminiferal communities, and the identification of key species in this process is essential to understand marine OM fluxes."

Added sentence: "In estuaries e.g. temperature acts in many cases as the most controlling factor on metabolic rates and hence on net ecosystem metabolism (Caffrey, 2003). Therefore, this factor was included in one of our observations concerning foraminiferal OM processing."

Materials and Methods R1: I think that the authors should provide more information regarding their experimental design. For example, for Experiment 1, why did they choose to terminate the incubation after 24 hours? Is this enough time to obtain a significant result?

JW: The short experimental period was chosen due to the following considerations: - To keep the effect of bacterial activity low. The foraminifera were cleaned before their transfer to the filtered incubation medium – but foraminiferal tests or cytoplasm always contain bacterial contaminations. Increased incubation time increases bacterial numbers and their contribution to the degradation of the algal material. Further, bacteria are incorporated together with the detrital diet. - The foraminifera were incubated in 6 well plates containing a volume of 12 mL NSW. A shorter incubation time assures the stability of the system. - To minimise potential stress due to laboratory cultivation in long-term incubations. A relatively high mortality was observed in earlier long term studies, specifically in A. tepida (Wukovits et al. 2017).

The results in Wukovits 2017 (carried out on individuals sampled in the same area)

further show, that time does not have a significant effect on the uptake of phytodetrital carbon in either of the two species after 2 days (in a time span of 2 - 14 days), suggesting that food intake and release equilibrates in a period prior to 2 days for these two intertidal species. Further, Moodley et al. (2000) observed a satiation of food intake in A. tepida within 50 hours after addition of phytodetritus in feeding experiments carried out on sediment cores.

The following sentences were added for clarity:

Line 137-138: "The experimental period of 24 hours was chosen to avoid potential bacterial activity and to maintain system stability."

R1: Why chlorophyte was not tested in Experiment 1?

JW: There is already a study, testing the feeding behaviour of the two species with a chlorophyte food source at different temperatures (Wukovits et al. 2017). Therefore, we focused on the diatom food source in this study.

R1: Why H. germanica was not included in Experiment 2?

JW: The sediment collected for Experiment 2 contained mainly A. tepida individuals (most likely due to a reproductive event shortly before the sampling date). Unfortunately, H. germanica individuals were not available in sufficient abundances to carry out a parallel run with this species. R1: Why was 20°C (and not 15°C or 25°C) the temperature tested in Experiment 2?

JW: Since A. tepida responses well to this temperature (Wukovits et al. 2018), 20°C was chosen. Temperatures in this range can further be measured in tide pools in the field in our sampling area in May/June.

The following sentence was added in the method description for Experiment 2: Line 145 – 146: " This experiment was carried out at 20°C, since A. tepida specimens collected in this area showed a good feeding response at this temperature (Wukovits et al., 2017).

R1: Line 88-90. How were these atom%s established?

JW: The atom%s of the final artificial phytodetritus were established by enriching the culture medium with aliquotes of NaH13CO3 and Na15NO3. The 13C labelling in D. tertiolecta in Experiment 2 was rather high (this complicates the IRMS-analysis), therefore, the 13C label addition was lowered for the production of the artificial phytodetritus in Experiment 1. (Experiment 2 was originally planned and carried out earlier than Experiment 1 (but there was not enough H. germanica material available to carry out a parallel with this species). But switching the sequence in the manuscript appeared to be more concise – first focusing on the comparison of the two species (since they are both mentioned in the title) and then going into more detail on the feeding preferences of one of the two species.)

The following section was added for more clarity about the algae cultivation methods and the establishment of the product's atom%: Line 117-123: "The algae culture medium for Experiment 1 (P. tricornutum) was produced with filtered NSW and enriched with 0.6 mM NaH13CO3 and 0.9 mM NaNO3 (Na14NO3 : Na15NO3 → 5.25 : 1), along with the stock solutions for the F/2 standard protocol. The culture medium for D. tertiolecta (13C single labeled) in Experiment 2 was produced with filtered NSW, the stock solutions for according to the F/2 standard protocol and additionally enriched with 1.5 mM NaH13CO3 and for P. tricornutum (15N single labelled) with 1.5 mM NaHCO3 (natural abundance) and with 0.9 mM NaNO3 (Na14NO3 : Na15NO3 → 5.25 : 1) along with the stock solutions for the F/2 standard protocol."

R1: Line 93. Is 28 PSU the same salinity as at the sampling site?

JW: The salinity range in our sampling underlies high seasonal and diurnal fluctuations depending on tidal activity, solar radiation, precipitation etc.. Our own measurements at the sampling site range between 24 PSU (water collected at high tide) and 31 PSU (water collected from a tidal pool at low tide). We completed the sentence: Line 131-132: "…which lies in the range of our measurements from seawater at the sampling

site: 24 – 30 PSU." Additional adjustment in the method section: in the new manuscript, North Sea seawater is abbreviated as NSW. (Line 109: "...filtered North Sea water (NSW)".)

R1: Lines 103-109 and 124-128. My suggestion is to explain the statistical treatment of the data in a separate section.

JW: The description of statistical treatment was transferred to a new section at the end of the Material and Methods section.

R1: Line 132: "The sediment core data, together with the data from laboratory experiments, were used to estimate (...)" The authors combined sediment core data with data from laboratory experiments to estimate total foraminiferal biomass and foraminiferal C and N processing. My question is why? The data obtained from the sediment core ("natural abundance") should be compared (and not combined) with the ones obtained from the laboratory experiments, as experiments are a simplification of the natural environment.

JW: The sentence was changed: Line 162-164: "The data from the laboratory experiments (individual TOC, TN, pC, pN), together with the foraminiferal abundances counted from the sediment core were used to estimate the range of foraminiferal contributions to sedimentary carbon and nitrogen pools and fluxes."

JW: An additional section was added to the discussion, were we discuss the importance of laboratory results to estimate ranges of foraminiferal contributions to carbon and nitrogen fluxes and pools. Line 314-334: " Our phytodetritus uptake estimates propose, that the foraminiferal biomass consists of $\sim 6-8\%$ diatom-derived pC /TOC, with the major amount contained within A. tepida (compare Table 3). An in-situ feeding experiment with deep-sea foraminifera resulted in values of $\sim 1-12\%$ pC/TOC (Nomaki et al., 2005b). Similar in-situ incubations in the core of the oxygen minimum zone of the Arabian Sea report $\sim 15\%$ pC/TOC in epifaunal and shallow infaunal foraminiferal carbon uptake (Enge et al., 2014). In-situ incubations offer results closest

to the natural responses of organisms in their natural habitat and enable precise estimates of foraminiferal nutrient fluxes. Although, specific microhabitat conditions can have a strong influence on organismic behaviour. The artificial conditions in laboratory experiments also have an influence on physiological analysis, therefore the obtained results should be treated with caution. However, our estimates lie in the same order of magnitude as the above mentioned in-situ studies and offer a basis for estimations on foraminiferal carbon and nitrogen fluxes. General variations in foraminiferal carbon and nitrogen budgets can be caused by different adaptations to variable food availability in different habitats. This can be achieved by different controls of energy metabolism (e.g. Linke, 1992) or different trophic strategies (e.g. Lopez, 1979; Nomaki et al., 2011; Pascal et al., 2008). Our results suggest, A. tepida has a higher relevance for intertidal OM processing than H. germanica. This can be mainly attributed to the sequestered chloroplasts within the cytoplasm of H. germanica. Kleptoplasty is a wide spread phenomenon in foraminifera, specifically in species inhabiting dysoxic sediments, where kleptoplasts could promote survival in anoxic pore waters (Bernhard and Bowser, 1999). They might be involved in biochemical pathways within the foraminiferal cytoplasm, e.g. the transport of inorganic carbon and nitrogen (LeKieffre et al., 2018). Further, transmission electron microscopic investigations on H. germanica report a very limited abundance of food vesicles (Goldstein and Richardson, 2018). Kleptoplast-bearing species might occupy a distinct niche concerning their energetic demands. Additionally, they might play a not yet discovered importance in the fluxes of inorganic or dissolved carbon and nitrogen compounds. However, secondary producers with high uptake rates and a quick response to particulate OM sources like A. tepida play a strong role in the biogeochemical carbon and nitrogen recycling."

R1: Line 140. After decalcification, the authors kept the foraminiferal at 50°C to dry for three days. Are the authors using a published protocol? If so, please cite the reference. If not, is it possible that such a long drying step could have altered their results? ˆ

JW: The drying step is critical in the processing of EA-IRMS samples. It is important,

that there is no moisture in the tin cups after complete decalcification (also, the tin cups containing the specimens have to be checked under the microscope to evaluate, if all individuals are on the bottom of the cup during/after addition of HCl to make sure that they are decalcified successfully). To our knowledge, drying at 50°C for 3 days does not alter TOC and TN, or 13C/12C and 15N/14N results, we used this method in many previous invetsigations (see added references below).

References to published protocol added: Line 172: "(Enge et al., 2014, 2016; Wukovits et al., 2017, 2018)"

R1: Table 1, 2nd column. "50 – 55". Are 50 the number of specimens used in the 24/fed experiment and 55 the number of specimens used in the 24/starved experiment? If so, please specify. [h] should be [hrs] for consistency with the rest of the manuscript.

JW: this was clarified in the text: Line 130: "Fifty to fifty five specimens of A. tepida and or H. germanica respectively..."

Results R1: I invite the authors to consider reporting the data presented in figure 1 as an additional (supplementary?) table.

JW: The raw data of the measurements for this study is available as a supplementary table in the revised manuscript.

R1: Figure 1 c and d. Considering that the temperature is specified in the x axis, I do not think that the authors need to colour code the data points, also because the "middle" shade of grey and the darker shade of grey cannot be easily distinguished. An alternative might be using different symbols for different temperatures. Also the meaning of "ns" is not included in the caption.

JW: The data points are now all coloured in black. The meaning of "ns" is now included in the caption. Line 193: "...food/24 hrs starved; p < 0.05, pairwise permutation tests, ns = not significant"

R1: Figure 2a. Can the data be differentiated based on the temperature of the experiments? Maybe different symbols (or colors) can be used for this purpose.

JW: A color code was added for the data points temperatures in Figure 2a and is shown in the legend of the figure.

R1: Figure 2b. The figure is a bit confusing. Again, I would recommend using different symbols (or colors) for different trends. JW: The figure was changed, now using different symbols for carbon and nitrogen release.

R1: Figure 3. Chlorphyte should Chlorophyte. Also not all symbols of the figure legend correspond to the symbols on the plots.

JW: "Chlorphyte" was changed in to "Chlorophyte". The figure was changed, the figure shows now uniform symbols which fit to the legend.

Discussion R1: The authors mention the presence of chloroplasts in Haynesina germanica. How about Ammonia tepida (cf. Jauffrais 2016).

JW: This is now already mentioned in the introduction of the revised mansucripte: Line 167-169: ". In contrary, food-derived chloroplasts in A. tepida lose their photosynthetic activity already within two days (Jauffrais et al., 2016)."

R1: Line 294-296. I think the authors make a very interesting point here. Can they expand on this?

JW: The last paragraph was rewritten: Line 356-367. " Therefore, foraminiferal nitrogen release as $NH_4+$ or amino-acids could cover a considerable amount of the nutritional nitrogen demand in marine bacteria (cf. Wheeler and Kirchman, 1986), which assimilate $NH_4+$ (and amino acid-derived $NH_4+$) to sustain their glutamate-glutamine cycle. Vice versa, the labile dissolved organic matter derived from bacterial decomposition of refractory organic matter provides a valuable food source for some benthic foraminifera, and is indispensable for the reproduction of some foraminiferal species (Jorissen et al., 1998; Muller and Lee, 1969; Nomaki et al., 2011). In many marine diatoms, which are the main drivers of marine primary productivity, $NH_4+$ is the preferred source for nitrogen uptake over NO3- (Sivasubramanian and Rao, 1988). Foraminifera could act as important nutrient providers for closely associated diatoms, which are also considered as one of their main food sources (Lee et al., 1966). Consequently, the kleptoplast-hosting metabolism in H. germanica could benefit from regenerated nitrogen sources by the high OM mineralization rates in A. tepida. In summary, foraminiferal carbon and nitrogen fluxes constitute an important link in the food web complex of primary consumers and decomposers.

Minor comments

R1: Line 12. Should '13C & 15N' be '13C & 15N'? This comment applies to the rest of the manuscript.

JW: 13C & 15N were substituted by 13C and 15N.

R1: Line 14-19. Throughout the mansuscripte, the results obtained in A. tepida are discussed before those obtained in H. germanica. I recommend maintaining the same structure in the abstract, as well.

JW: The sequence in the abstract was changed: Line 13 – 21: "Ammonia tepida showed a very high, temperature-influenced intake and turnover rates with more excessive carbon turnover, compared to nitrogen. The quite low metabolic nitrogen turnover in H. germanica was not affected by temperature and was higher than the carbon turnover. This might be related with the chloroplast husbandry in H. germanica and its lower demands for food derived nitrogen sources. Ammonia tepida prefers a soft chlorophyte food source over diatom detritus, which is harder to break down. In conclusion, A. tepida shows a generalist behaviour that links with high fluxes of organic matter (OM). Due to its high rates of OM processing and abundances, we conclude that A. tepida is an important key-player in intertidal carbon and nitrogen turnover, specifically in the short-term processing of OM and the mediation of dissolved nutrients to associated microbes and primary producers. In contrast, H. germanica is a highly specialized species with low rates of carbon and nitrogen budgeting."

R1: Line 25: "Coastal sediments represent the largest pool of marine particulate organic matter (OM)...' Can the authors add some numbers (maybe a percentage?) regarding how big the OM pool is in coastal sediments? In my opinion, such a number will provide a good context to discuss the data obtained from the experiments and to discuss the importance of remineralization processes mediated by benthic foraminifera in coastal environments.

JW: The following sections have been added: Line 24-31: "Oceanic and terrestrial systems are connected by the carbon cycling in coastal waters, which contribute to a major part of the global carbon cycles and budgets (Bauer et al., 2013; Cai, 2011; Cole et al., 2007; Regnier et al., 2013). Estuaries are an important source for organic matter in coastal systems and were estimated to account for $\sim$ 40% of oceanic phytoplankton primary productivity (Smith and Hollibaugh 1993). Most estuarine areas are considered to be net heterotrophic, or act as carbon sinks, respectively (e.g. Caffrey, 2003, 2004; Cai, 2011; Herrmann et al., 2015). In general, 30% of overall coastal carbon is lost by metabolic oxidation (Smith and Hollibaugh 1993)."

Line 336-341: "As mentioned above, in the heterotrophic, coastal zone 30% of the carbon pool are lost as via respiration. On the other hand, dissolved organic carbon sources from organismic excretion can serve as an important nutrient source for bacteria (e.g., Kahler et al., 1997; Snyder & Hoch, 1996; Zweifel et al., 1993). Therefore, the fast processing of OM in A. tepida might be an important sink for inorganic carbon ($CO_2$ respiration) and at the same time a link for dissolved organic carbon sources in intertidal carbon and nitrogen fluxes."

R1: Line 36: "e.g., temperature or OM quality". This should be "temperature and/or OM quality".

JW: this was changed according to the reviewers suggestion.

R1: Lines 40-41 and 47-48. These sentences are not very clear. Please rephrase.

JW: these sentences were rephrased as follows: Line 49-53: "Typically, tidal flats offer a high availability of food sources for phytodetrivores or herbivores feeding on microalgae. But dense populations of A. tepida communities can deplete sediments from OM sources and consequently control benthic meiofaunal community structures (Chandler, 1989). Therefore, resource partitioning or different metabolic strategies can be beneficial for foraminifera which share the same spatial and temporal habitats." Line 82-85: "Therefore, seasonal temperature fluctuations and human induced global warming can have a strong impact on foraminiferal community compositions and foraminiferal C & N fluxes." 2 further sentences were added: "In estuaries e.g. temperature acts in many cases as the most controlling factor on metabolic rates and hence on net ecosystem metabolism (Caffrey, 2003). Therefore, this factor was included in one of our observations concerning foraminiferal OM processing."

R1: Lines 58-59. Considering that the experiment described at lines 58-59 is Experiment #2, I suggest moving this sentence after the sentence at lines 60-61, which refers to Experiment #1.

JW: This shift was done: Line 90-94: "We compared diatom detritus intake and retention of food-derived carbon (pC) and nitrogen (pN) of A. tepida and H. germanica at three different temperatures (15°C, 20°C, 25°C). The evaluation of the metabolic costs of pC and pN during a 24 hour starvation period can further help to explain species specific OM processing due to metabolic nutrient budgets. Further, both food sources were offered simultaneously to A. tepida to identify feeding preferences of this species."

R1: M2 should be m2

JW: replaced with m2

R1: "Individuals were picked from the sediment in sufficient and collected (...)". In sufficient number?

JW: Yes, sentence was completed: Line 108: "Foraminifera were picked from the sediment in sufficient number and collected (...)".

R1: Line 77: "Dunaliella tertiolecta and Phaeodactylum tricornutum". The scientific name was already defined at line 58, so this should be D. tertiolecta and P. tricornutum. This comment applies to the rest of the manuscript, with the exception of tables and figures.

JW: These changes were carried out.

R1: "The experiments started after accumulation of sufficient foraminiferal material three weeks after the field sampling." I assume the authors achieved foraminiferal reproduction during the initial incubation. If my assumption is correct, then it would be good to specify so and provide some information about the conditions used to maintain the foraminifera prior the beginning of the experiments. If the authors know, it might be of interest to know how successful the reproduction event was.

JW: Upon arrival at the lab, the sediment was immediately transferred into aerated aquaria containing filtered seawater at the sampling site. We did not monitor reproduction during the incubation period. The following sentence was added to the revised manuscript: Line 107-108: "The sediment samples were kept within aquaria, containing filtered water collected at the sampling site."

R1: Line 84. NaH13CO3, Na15NO3 should be NaH13CO3, Na15NO3.

JW: changed.

R1: Line 88. C.f. should be cf. This comment applies to the rest of the manuscript.

JW: changed.

R1: Line 108. What do the authors mean with "carbon and nitrogen costs of the two species during the period without food"?

JW: sentecne changed: Line 196-197: "...metabolic carbon and nitrogen loss of the two species during the period without food."

R1: Line 114. Cm-2 should be cm-2.

JW: changed.

R1: Line 135. A parenthesis is missing.

JW: Parenthesis added.

R1: Line 137. I suggest including the word "cytoplasm" prior "isotope analysis", for clarity.

JW: The word "cytoplasm" was included.

R1: Line 153 (formula #2). atomXsample – should this be atom%Xsample? Same for background.

JW: "atomXsample" was replaced by "atom%Xsample" in both cases.

R1: Line 155. I recommend writing the Iiso formula as the other formulas, for clarity.

JW: The Iiso formula was written as the other formulas.

R1: Line 155. There is an extra period after Table 2.

JW: Extra period removed.

R1: Line 205. No comma needed.

JW: Comma removed.

R1: Line 212. Phaeodactylum tricornutum should be italic.

JW: Phaeodactylum tricornutum was changed to "P. tricornutum".

R1: Section 4.1 revise references – e.g., a comma is missing between the authors' names and the year of publication and a semicolon should be used to separate different references.

JW: The reference style was adapted to biogeosciences.

R1: Line 232. Missing parenthesis.

JW: Parenthesis added.

R1: Line 250. Almagor et al. – publication year 1981.

JW: Publication year added.

R1: Lines 281 and 295. Missing parenthesis around the year of publication. JW: Parenthesis added.

R1: Lines 288 and 295. Comp. should be probabyl cf.

JW: Comp. replaced by cf.

Please also note the supplement to this comment:
https://www.biogeosciences-discuss.net/bg-2018-231/bg-2018-231-AC1-
supplement.pdf

**Supplement:**

[revised manuscript text omitted]

Supplementary Figure 1. a) Light microscope image of fresh picked *A. tepida* specimens (scale bar = 500 µm). b) *A. tepida* after feeding on fresh microalgae. c) Fresh picked *H. germanica* specimens (scale bar = 500 µm). d) *H. germanica* individual (scale bar = 200 µm). e)-h) SEM images of *A. tepida* collected in 2014 at the sampling location of this study (scale bar = 200 µm). i)-j) *H. germanica* collected in 2014 at the sampling location of this study (scale bar = 200 µm). k)-l) *A. tepida* collected in 2016 at the sampling location of this study (scale bar = 200 µm). m) *H. germanica* collected in 2016 at the sampling location of this study (scale bar = 200 µm).

Table **S1**. Raw data of EA-IRMA of foraminiferal samples of Paper 4 (n.a. = natural abundance, data for 5.5 P from Paper 3).

|  | d 15N/14N | AT% 15N/14N | d 13C/12C | AT% 13C/12C |
|---|---|---|---|---|
| *H. germanica* n.a. | 12.58 | 0.371 | -13.56 | 1.091 |
| 5.7 D | 45528.61 | 14.613 | 1153.00 | 2.351 |
| 7.1 D | 216695.58 | 44.464 | 10361.59 | 11.271 |
| 5.5 P | 32011.61 | 15.824 | 1627.25 | 4.389 |

| food source | T | treatment | Nr/Ind | weight [mg] | d 15N/14N | AT% 15N/14N | d 13C/12C | AT% 13C/12C | µg N | µg C |
|---|---|---|---|---|---|---|---|---|---|---|
| 5.7 D | 15°C | 24 hrs fed | 48 | 1.250 | 425.11 | 0.521 | 3.82 | 1.110 | 8.85 | 51.58 |
| 5.7 D | 15°C | 24 hrs fed | 52 | 1.277 | 416.26 | 0.518 | 3.25 | 1.109 | 8.79 | 52.80 |
| 5.7 D | 15°C | 24 hrs fed | 53 | 1.289 | - | - | - | - | - | - |
| 5.7 D | 20°C | 24 hrs fed | 50 | 0.963 | 328.42 | 0.486 | -0.42 | 1.105 | 7.75 | 43.83 |
| 5.7 D | 20°C | 24 hrs fed | 53 | 1.284 | 323.13 | 0.484 | -0.54 | 1.105 | 8.61 | 51.50 |
| 5.7 D | 20°C | 24 hrs fed | 56 | 1.353 | 335.27 | 0.489 | 0.39 | 1.106 | 8.74 | 52.34 |
| 5.7 D | 25°C | 24 hrs fed | 48 | 0.972 | 257.92 | 0.461 | -2.90 | 1.102 | 8.85 | 49.25 |
| 5.7 D | 25°C | 24 hrs fed | 54 | 1.301 | 291.73 | 0.473 | -3.30 | 1.102 | 9.19 | 53.82 |
| 5.7 D | 25°C | 24 hrs fed | 50 | 1.176 | 385.61 | 0.507 | 2.53 | 1.108 | 7.89 | 45.38 |
| 5.7 D | 15°C | 24 hrs starved | 49 | 1.052 | 230.04 | 0.450 | -4.70 | 1.101 | 7.58 | 42.71 |
| 5.7 D | 15°C | 24 hrs starved | 61 | - | - | - | - | - | - | - |
| 5.7 D | 15°C | 24 hrs starved | 53 | 1.192 | 211.71 | 0.444 | -5.63 | 1.099 | 8.70 | 51.81 |
| 5.7 D | 20°C | 24 hrs starved | 48 | 1.108 | 241.56 | 0.455 | -5.04 | 1.100 | 7.44 | 41.11 |
| 5.7 D | 20°C | 24 hrs starved | 45 | 1.099 | 254.99 | 0.459 | -5.81 | 1.099 | 7.96 | 43.16 |
| 5.7 D | 20°C | 24 hrs starved | 44 | 1.202 | 234.45 | 0.452 | -6.98 | 1.098 | 7.39 | 42.01 |
| 5.7 D | 25°C | 24 hrs starved | 49 | 1.217 | 337.63 | 0.490 | -3.98 | 1.101 | 7.70 | 39.16 |
| 5.7 D | 25°C | 24 hrs starved | 39 | 1.057 | 321.86 | 0.484 | -4.15 | 1.101 | 7.18 | 40.77 |
| 5.7 D | 25°C | 24 hrs starved | 43 | 1.235 | 374.66 | 0.503 | -2.93 | 1.102 | 6.66 | 36.67 |
| 7.1 D | 15°C | 24 hrs fed | 55 | 1.411 | 496.43 | 0.547 | 35.23 | 1.144 | 9.48 | 55.00 |
| 7.1 D | 15°C | 24 hrs fed | 52 | 1.158 | 619.09 | 0.592 | 50.35 | 1.161 | 9.60 | 57.15 |
| 7.1 D | 15°C | 24 hrs fed | 53 | 1.417 | 481.34 | 0.542 | 41.18 | 1.151 | 8.80 | 51.30 |
| 7.1 D | 20°C | 24 hrs fed | 49 | 1.373 | 1004.73 | 0.732 | 94.37 | 1.209 | 9.01 | 50.38 |
| 7.1 D | 20°C | 24 hrs fed | 50 | 0.907 | 1148.04 | 0.784 | 113.35 | 1.229 | 7.48 | 42.49 |
| 7.1 D | 20°C | 24 hrs fed | 49 | 1.413 | 1017.81 | 0.737 | 108.25 | 1.224 | 9.49 | 54.69 |
| 7.1 D | 25°C | 24 hrs fed | 51 | 1.243 | 853.93 | 0.677 | 80.93 | 1.194 | 8.90 | 48.30 |
| 7.1 D | 25°C | 24 hrs fed | 48 | 1.195 | 934.13 | 0.706 | 83.27 | 1.197 | 7.60 | 43.78 |
| 7.1 D | 25°C | 24 hrs fed | 53 | 1.243 | 764.11 | 0.645 | 65.33 | 1.177 | 8.23 | 46.30 |
| 7.1 D | 15°C | 24 hrs starved | 50 | 1.128 | 432.69 | 0.524 | 12.73 | 1.120 | 7.28 | 40.52 |
| 7.1 D | 15°C | 24 hrs starved | 46 | 1.166 | 465.91 | 0.536 | 16.65 | 1.124 | 7.27 | 43.98 |
| 7.1 D | 15°C | 24 hrs starved | 59 | 1.251 | 416.39 | 0.518 | 15.44 | 1.123 | 8.80 | 51.35 |
| 7.1 D | 20°C | 24 hrs starved | 49 | 1.160 | 555.79 | 0.569 | 23.69 | 1.132 | 7.88 | 44.51 |
| 7.1 D | 20°C | 24 hrs starved | 48 | 1.209 | 715.38 | 0.627 | 35.48 | 1.144 | 7.74 | 42.10 |
| 7.1 D | 20°C | 24 hrs starved | 50 | 1.280 | 771.33 | 0.647 | 34.82 | 1.144 | 8.32 | 45.05 |
| 7.1 D | 25°C | 24 hrs starved | 53 | 1.335 | 797.43 | 0.657 | 33.54 | 1.142 | 8.08 | 45.90 |
| 7.1 D | 25°C | 24 hrs starved | 48 | 1.144 | 855.89 | 0.678 | 51.27 | 1.162 | 7.87 | 41.64 |
| 7.1 D | 25°C | 24 hrs starved | 54 | 1.230 | 845.19 | 0.674 | 43.79 | 1.154 | 8.00 | 45.05 |
| 5.5 P | 15°C | 24 hrs fed | 50 | 1.469 | 541.16 | 0.564 | 64.97 | 1.177 | 8.27 | 49.43 |
| 5.5 P | 15°C | 24 hrs fed | 51 | 1.359 | 618.81 | 0.592 | 68.13 | 1.180 | 9.73 | 56.12 |
| 5.5 P | 15°C | 24 hrs fed | 51 | 1.398 | 559.13 | 0.570 | 60.14 | 1.171 | 9.53 | 54.99 |
| 5.5 P | 20°C | 24 hrs fed | 52 | 1.172 | 598.98 | 0.585 | 58.26 | 1.169 | 10.87 | 60.65 |
| 5.5 P | 20°C | 24 hrs fed | 52 | 1.363 | 576.87 | 0.577 | 62.15 | 1.174 | 10.05 | 52.00 |
| 5.5 P | 20°C | 24 hrs fed | 51 | 1.117 | 484.73 | 0.543 | 57.47 | 1.168 | 10.69 | 53.97 |
| 5.5 P | 25°C | 24 hrs fed | 47 | 1.012 | 439.52 | 0.527 | 32.98 | 1.142 | 9.38 | 46.52 |
| 5.5 P | 25°C | 24 hrs fed | 49 | 1.114 | 461.60 | 0.535 | 33.99 | 1.143 | 9.49 | 46.77 |
| 5.5 P | 25°C | 24 hrs fed | 49 | 1.127 | 403.72 | 0.514 | 34.21 | 1.143 | 10.93 | 54.56 |
| 5.5 P | 15°C | 24 hrs starved | 48 | 1.246 | 364.87 | 0.500 | 44.50 | 1.154 | 7.43 | 40.48 |
| 5.5 P | 15°C | 24 hrs starved | 52 | 1.531 | 397.28 | 0.511 | 47.21 | 1.157 | 9.18 | 52.64 |
| 5.5 P | 15°C | 24 hrs starved | 53 | 1.508 | 351.37 | 0.495 | 48.17 | 1.158 | 9.13 | 50.88 |
| 5.5 P | 20°C | 24 hrs starved | 51 | 1.098 | 395.17 | 0.511 | 43.51 | 1.153 | 10.00 | 49.00 |
| 5.5 P | 20°C | 24 hrs starved | 54 | 1.019 | 365.61 | 0.500 | 40.76 | 1.150 | 9.27 | 45.91 |
| 5.5 P | 20°C | 24 hrs starved | 55 | 1.322 | 362.58 | 0.499 | 31.43 | 1.140 | 9.90 | 48.50 |
| 5.5 P | 25°C | 24 hrs starved | 54 | 1.046 | 289.12 | 0.472 | 18.97 | 1.126 | 9.62 | 50.96 |
| 5.5 P | 25°C | 24 hrs starved | 39 | 0.952 | 333.26 | 0.488 | 22.21 | 1.130 | 7.14 | 36.69 |
| 5.5 P | 25°C | 24 hrs starved | 57 | 1.327 | 313.36 | 0.481 | 19.12 | 1.127 | 10.97 | 52.50 |

Table **S2**. Raw data of GC-IRMS of water samples.

| | food source | T | | avg d13C/12C | avg AT% 13C/12C | ppm CO2 |
|---|---|---|---|---|---|---|
| H₃PO₄ | | - | - | -18.734 | 1.0852 | 83.85 |
| H₃PO₄ + SW | | - | - | -1.537 | 1.1040 | 9162.91 |
| | 5.7 D | 15°C | | -1.135 | 1.1044 | 9398.27 |
| | 5.7 D | 15°C | | -1.250 | 1.1043 | 8406.95 |
| | 5.7 D | 15°C | | -1.156 | 1.1044 | 9883.24 |
| | 5.7 D | 20°C | | -0.870 | 1.1047 | 9184.63 |
| | 5.7 D | 20°C | | -1.162 | 1.1044 | 7778.55 |
| | 5.7 D | 20°C | | -0.994 | 1.1046 | 8489.17 |
| | 5.7 D | 25°C | | -0.963 | 1.1046 | 8829.57 |
| | 5.7 D | 25°C | | -1.047 | 1.1045 | 9614.38 |
| | 7.1 D | 15°C | | -0.887 | 1.1047 | 9472.19 |
| | 7.1 D | 15°C | | -0.756 | 1.1048 | 9247.92 |
| | 7.1 D | 15°C | | -1.060 | 1.1045 | 9794.93 |
| | 7.1 D | 20°C | | -0.941 | 1.1046 | 6902.25 |
| | 7.1 D | 20°C | | -0.762 | 1.1048 | 7843.66 |
| | 7.1 D | 20°C | | -0.910 | 1.1047 | 7959.04 |
| | 7.1 D | 25°C | | -0.306 | 1.1053 | 7818.50 |
| | 7.1 D | 25°C | | -0.465 | 1.1052 | 9206.42 |
| | 5.7 P | 15°C | | -0.861 | 1.1047 | 7659.74 |
| | 5.7 P | 15°C | | -0.971 | 1.1046 | 9208.57 |
| | 5.7 P | 15°C | | -0.655 | 1.1049 | 9055.14 |
| | 5.7 P | 20°C | | -0.536 | 1.1051 | 9910.84 |
| | 5.7 P | 20°C | | -0.538 | 1.1051 | 8473.16 |
| | 5.7 P | 20°C | | -0.691 | 1.1049 | 9219.05 |
| | 5.7 P | 25°C | | -0.351 | 1.1053 | 7268.07 |
| | 5.7 P | 25°C | | -0.664 | 1.1049 | 7246.05 |
| He | | | | | | 0 |

Response to Referee 1:

**Major comments**

Introduction

R1: Introduction. I think that the manuscript would benefit from a greater overview of: 1) the carbon and nitrogen cycles in coastal environments 2) the role of benthic foraminifera in the carbon and nitrogen cycles. Some information are provided in the Discussion section of the Manuscript. However, I think that a general overview of these porcesses should be included in the Introduction, as well.

> JW: The introduction was extended, providing the following information about coastal carbon and nitrogen cycles and the role of foraminifera in these cycles:

> Line 26-43: "Oceanic and terrestrial systems are connected by the carbon cycling in coastal waters, which contribute to a major part of the global carbon cycles and budgets (Bauer et al., 2013; Cai, 2011; Cole et al., 2007; Regnier et al., 2013). Estuaries are an important source for organic matter in coastal systems and were estimated to account for ~ 40% of oceanic phytoplankton primary productivity (Smith and Hollibaugh 1993). Most estuarine areas are considered to be net heterotrophic, or act as carbon sinks, respectively (e.g. Caffrey, 2003, 2004; Cai, 2011; Herrmann et al., 2015). In general, 30% of overall coastal carbon is lost by metabolic oxidation (Smith and Hollibaugh 1993). Foraminifera are highly abundant in estuarine sediments and contribute strongly to these processes (Alve and Murray, 1994; Cesbron et al., 2016; Moodley et al., 2000; Murray and Alve, 2000). They feed on various sources of labile particulate OM, including microalgae and detritus, and provide a pivotal link in marine carbon cycles and food webs (Bradshaw, 1961; Goldstein and Corliss, 1994; Heinz et al., 2001; Lee et al., 1966; Lee and Muller, 1973; Nomaki et al., 2005b, 2006, 2009, 2011). The nitrogen compounds of OM particles are usually remineralized to ammonium ($NH4+$). In this way, nitrogen gets again available as nutrient for primary productivity. A major part of this process is attributed to prokaryotic degraders, but protists are also involved in the process of regeneration of organic nitrogen compounds (Ferrier-Pages and Rassoulzadegan, 1994; Ota and Taniguchi, 2003; Verity et al., 1992). Due to their high abundances, we consider, that foraminifera contribute a large part to this OM reworking and the regeneration of carbon and nitrogen compounds from particulate OM sources, e.g. phytodetritus. In this study, we quantify the bulk OM-derived carbon and nitrogen release, which originates rather via excretion of organic carbon and nitrogen compounds (vesicular transport of metabolic waste products), respiration or diffusion of inorganic carbon and nitrogen by these single celled microorganisms."

R1: Line 49: The authors briefly mention previous studies on feeding preferences/strategy. Considering that these are important points that are discussed later in the manuscript, I suggest providing more information regarding past experimental studies. In doing so, the authors can better emphasize the novelty of their work in the context of earlier investigations.

> JW: Added section:

> Line 70-79: „Laboratory feeding experiments have shown, that *A. tepida* responds to several food sources, including different live microalgae (chlorophytes and diatoms) and chlorophyte

and diatom detritus (Bradshaw, 1961; Lee et al., 1966; LeKieffre et al., 2017; Linshy et al., 2014; Pascal et al., 2008; Wukovits et al., 2017, 2018). On the other hand, *H. germanica* shows a low affinity to chloroplast detritus food sources (Wukovits et al., 2017), but feeds actively on diatoms (Ward et al., 2003) and takes up inorganic, dissolved C & N compounds (LeKieffre et al., 2018). Both species are found in muddy coastal sediments containing high loads of nutrients or OM (Armynot du Chatelet et al., 2009; Armynot du Châtelet et al., 2004). But considering their different feeding strategies they might play distinct roles in the reworking of OM. Recent literature still lacks direct, quantitative comparisons of foraminiferal species-specific quantitative OM-derived C & N ingestion and release. Therefore, this study aims to compare and quantify variations in their respective uptake of OM (phytodetritus).

R1: Line 56. This work might be of interest to readers who might not be familiar with foraminifera. Thus, I recommend to better explaining what the authors mean by „release of OM derived carbon and nitrogen in foraminifera" and how this connects with OM remineralization processes in coastal waters.

>JW: The following sentence was added:

>Line 41-43: „In this study, we quantify the bulk OM-derived C & N release, which originates rather via excretion of organic carbon and nitrogen compounds (vesicular transport of metabolic waste products), respiration or diffusion of inorganic C and N by the single celled micro-organisms."

>Additional changes in the introduction to better integrate the reviewers suggestions:

>The following section was removed to keep the introduction concise:

>"Certain key species in foraminiferal communities contribute with a major extant to the OM processing in extensive, highly productive marine environments (Enge et al., 2014, 2016, Moodley et al., 2000, 2002, Nomaki et al., 2005a, 2008; Witte et al., 2003; Wukovits et al., 2018). Therefore, the quantification of foraminiferal carbon and nitrogen processing derived from OM and food selectivity in foraminiferal communities, and the identification of key species in this process is essential to understand marine OM fluxes."

>Added sentence:

>„In estuaries e.g. temperature acts in many cases as the most controlling factor on metabolic rates and hence on net ecosystem metabolism (Caffrey, 2003). Therefore, this factor was included in one of our observations concerning foraminiferal OM processing."

**Materials and Methods**

R1: I think that the authors should provide more information regarding their experimental design. For example, for Experiment 1, why did they choose to terminate the incubation after 24 hours? Is this enough time to obtain a significant result?

>JW: The short experimental period was chosen due to the following considerations:

- To keep the effect of bacterial activity low. The foraminifera were cleaned before their transfer to the filtered incubation medium – but foraminiferal tests or cytoplasm always contain bacterial contaminations. Increased incubation time increases bacterial numbers and their contribution to the degradation of the algal material. Further, bacteria are incorporated together with the detrital diet.
- The foraminifera were incubated in 6 well plates containing a volume of 12 mL NSW. A shorter incubation time assures the stability of the system.
- To minimise potential stress due to laboratory cultivation in long-term incubations. A relatively high mortality was observed in earlier long term studies, specifically in *A. tepida* (Wukovits et al. 2017).
- The results in Wukovits 2017 (carried out on individuals sampled in the same area) further show, that time does not have a significant effect on the uptake of phytodetrital carbon in either of the two species after 2 days (in a time span of 2 - 14 days), suggesting that food intake and release equilibrates in a period prior to 2 days for these two intertidal species. Further, Moodley et al. (2000) observed a satiation of food intake in *A. tepida* within 50 hours after addition of phytodetritus in feeding experiments carried out on sediment cores.

The following sentences were added for clarity:

Line 137-138: „The experimental period of 24 hours was chosen to avoid potential bacterial activity and to maintain system stability.“

R1: Why chlorophyte was not tested in Experiment 1?

JW: There is already a study, testing the feeding behaviour of the two species with a chlorophyte food source at different temperatures (Wukovits et al. 2017). Therefore, we focused on the diatom food source in this study.

R1: Why *H. germanica* was not included in Experiment 2?

JW: The sediment collected for Experiment 2 contained mainly *A. tepida* individuals (most likely due to a reproductive event shortly before the sampling date). Unfortunately, *H. germanica* individuals were not available in sufficient abundances to carry out a parallel run with this species.

R1: Why was 20°C (and not 15°C or 25°C) the temperature tested in Experiment 2?

JW: Since *A. tepida* responses well to this temperature (Wukovits et al. 2018), 20°C was chosen. Temperatures in this range can further be measured in tide pools in the field in our sampling area in May/June.

The following sentence was added in the method description for Experiment 2:

Line 145 – 146: „ This experiment was carried out at 20°C, since *A. tepida* specimens collected in this area showed a good feeding response at this temperature (Wukovits et al., 2017).

R1: Line 88-90. How were these atom%s established?

JW: The atom%s of the final artificial phytodetritus were established by enriching the culture medium with aliquotes of $NaH^{13}CO_3$ and $Na^{15}NO_3$.

The $^{13}C$ labelling in *D. tertiolecta* in Experiment 2 was rather high (this complicates the IRMS-analysis), therefore, the $^{13}C$ label addition was lowered for the production of the artificial phytodetritus in Experiment 1.

(Experiment 2 was originally planned and carried out earlier than Experiment 1 (but there was not enough *H. germanica* material available to carry out a parallel with this species). But switching the sequence in the manuscript appeared to be more concise – first focusing on the comparison of the two species (since they are both mentioned in the title) and then going into more detail on the feeding preferences of one of the two species.)

The following section was added for more clarity about the algae cultivation methods and the establishment of the product's atom%:

Line 117-123: „The algae culture medium for Experiment 1 (*P. tricornutum*) was produced with filtered NSW and enriched with 0.6 mM $NaH^{13}CO_3$ and 0.9 mM $NaNO_3$ ($Na^{14}NO_3 : Na^{15}NO_3 \rightarrow$ 5.25 : 1), along with the stock solutions for the F/2 standard protocol. The culture medium for *D. tertiolecta* ($^{13}C$ single labeled) in Experiment 2 was produced with filtered NSW, the stock solutions for according to the F/2 standard protocol and additionally enriched with 1.5 mM $NaH^{13}CO_3$ and for *P. tricornutum* ($^{15}N$ single labelled) with 1.5 mM $NaHCO_3$ (natural abundance) and with 0.9 mM $NaNO_3$ ($Na^{14}NO_3 : Na^{15}NO_3 \rightarrow$ 5.25 : 1) along with the stock solutions for the F/2 standard protocol."

R1: Line 93. Is 28 PSU the same salinity as at the sampling site?

JW: The salinity range in our sampling underlies high seasonal and diurnal fluctuations depending on tidal activity, solar radiation, precipitation etc.. Our own measurements at the sampling site range between 24 PSU (water collected at high tide) and 31 PSU (water collected from a tidal pool at low tide).

We completed the sentence:

Line 131-132: "…which lies in the range of our measurements from seawater at the sampling site: 24 – 30 PSU."

Additional adjustment in the method section: in the new manuscript, North Sea seawater is abbreviated as NSW. (Line 109: „...filtered North Sea water (NSW)".)

R1: Lines 103-109 and 124-128. My suggestion is to explain the statistical treatment of the data in a separate section.

JW: The description of statistical treatment was transferred to a new section at the end of the Material and Methods section.

R1: Line 132: „The sediment core data, together with the data from laboratory experiments, were used to estimate (...)" The authors combined sediment core data with data from laboratory experiments to estimate total foraminiferal biomass and foraminiferal C and N processing. My question is why? The data obtained from the sediment core („natural abundance") should be compared (and not combined) with the ones obtained from the laboratory experiments, as experiments are a simplification of the natural environment.

JW: The sentence was changed:

Line 162-164: „The data from the laboratory experiments (individual TOC, TN, pC, pN), together with the foraminiferal abundances counted from the sediment core were used to estimate the range of foraminiferal contributions to sedimentary carbon and nitrogen pools and fluxes."

JW: An additional section was added to the discussion, were we discuss the importance of laboratory results to estimate ranges of foraminiferal contributions to carbon and nitrogen fluxes and pools.

Line 314-334: „ Our phytodetritus uptake estimates propose, that the foraminiferal biomass consists of ~ 6 – 8% diatom-derived pC /TOC, with the major amount contained within *A. tepida* (compare Table 3). An *in-situ* feeding experiment with deep-sea foraminifera resulted in values of ~ 1 – 12% pC/TOC (Nomaki et al., 2005b). Similar *in-situ* incubations in the core of the oxygen minimum zone of the Arabian Sea report ~ 15% pC/TOC in epifaunal and shallow infaunal foraminiferal carbon uptake (Enge et al., 2014). *In-situ* incubations offer results closest to the natural responses of organisms in their natural habitat and enable precise estimates of foraminiferal nutrient fluxes. Although, specific microhabitat conditions can have a strong influence on organismic behaviour. The artificial conditions in laboratory experiments also have an influence on physiological analysis, therefore the obtained results should be treated with caution. However, our estimates lie in the same order of magnitude as the above mentioned *in-situ* studies and offer a basis for estimations on foraminiferal carbon and nitrogen fluxes. General variations in foraminiferal carbon and nitrogen budgets can be caused by different adaptations to variable food availability in different habitats. This can be achieved by different controls of energy metabolism (e.g. Linke, 1992) or different trophic strategies (e.g. Lopez, 1979; Nomaki et al., 2011; Pascal et al., 2008). Our results suggest, *A. tepida* has a higher relevance for intertidal OM processing than *H. germanica*. This can be mainly attributed to the sequestered chloroplasts within the cytoplasm of *H. germanica*. Kleptoplasty is a wide spread phenomenon in foraminifera, specifically in species inhabiting dysoxic sediments, where kleptoplasts could promote survival in anoxic pore waters (Bernhard and Bowser, 1999). They might be involved in biochemical pathways within the foraminiferal cytoplasm, e.g. the transport of inorganic carbon and nitrogen (LeKieffre et al., 2018). Further, transmission electron microscopic investigations on *H. germanica* report a very limited abundance of food vesicles (Goldstein and Richardson, 2018). Kleptoplast-bearing species might occupy a distinct niche concerning their energetic demands. Additionally, they might play a not yet discovered importance in the fluxes of inorganic or dissolved carbon and nitrogen compounds. However, secondary producers with high uptake rates and a quick response to particulate OM sources like *A. tepida* play a strong role in the biogeochemical carbon and nitrogen recycling."

R1: Line 140. After decalcification, the authors kept the foraminiferal at 50°C to dry for three days. Are the authors using a published protocol? If so, please cite the reference. If not, is it possible that such a long drying step could have altered their results?          ^

> JW: The drying step is critical in the processing of EA-IRMS samples. It is important, that there is no moisture in the tin cups after complete decalcification (also, the tin cups containing the specimens have to be checked under the microscope to evaluate, if all individuals are on the bottom of the cup during/after addition of HCl to make sure that they are decalcified successfully). To our knowledge, drying at 50°C for 3 days does not alter TOC and TN, or 13C/12C and 15N/14N results, we used this method in many previous invetsigations (see added references below).

> References to published protocol added:

> Line 172: „(Enge et al., 2014, 2016; Wukovits et al., 2017, 2018)"

R1: Table 1, 2nd column. „50 – 55". Are 50 the number of specimens used in the 24/fed experiment and 55 the number of specimens used in the 24/starved experiment? If so, please specify. [h] should be [hrs] for consistency with the rest of the manuscript.

> JW: this was clarified in the text:

> Line 130: „Fifty to fifty five specimens of *A. tepida* and or *H. germanica* respectively…"

Results

R1: I invite the authors to consider reporting the data presented in figure 1 as an additional (supplementary?) table.

> JW: The raw data of the measurements for this study is available as a supplementary table in the revised manuscript.

R1: Figure 1 c and d. Considering that the temperature is specified in the x axis, I do not think that the authors need to colour code the data points, also because the „middle" shade of grey and the darker shade of grey cannot be easily distinguished. An alternative might be using different symbols for different temperatures. Also the meaning of „ns" is not included in the caption.

> JW: The data points are now all coloured in black. The meaning of „ns" is now included in the caption.

> Line 193: „...food/24 hrs starved; $p < 0.05$, pairwise permutation tests, ns = not significant"

R1: Figure 2a. Can the data be differentiated based on the temperature of the experiments? Maybe different symbols (or colors) can be used for this purpose.

> JW: A color code was added for the data points temperatures in Figure 2a and is shown in the legend of the figure.

R1: Figure 2b. The figure is a bit confusing. Again, I would recommend using different symbols (or colors) for different trends.

> JW: The figure was changed, now using different symbols for carbon and nitrogen release.

R1: Figure 3. Chlorphyte should Chlorophyte. Also not all symbols of the figure legend correspond to the symbols on the plots.

> JW: "Chlorphyte" was changed in to "Chlorophyte". The figure was changed, the figure shows now uniform symbols which fit to the legend.

Discussion

R1: The authors mention the presence of chloroplasts in *Haynesina germanica*. How about *Ammonia tepida* (cf. Jauffrais 2016).

> JW: This is now already mentioned in the introduction of the revised mansucripte:
>
> Line 167-169: „. In contrary, food-derived chloroplasts in *A. tepida* lose their photosynthetic activity already within two days (Jauffrais et al., 2016)."

R1: Line 294-296. I think the authors make a very interesting point here. Can they expand on this?

> JW: The last paragraph was rewritten:
>
> Line 356-367. „ Therefore, foraminiferal nitrogen release as $NH_4^+$ or amino-acids could cover a considerable amount of the nutritional nitrogen demand in marine bacteria (cf. Wheeler and Kirchman, 1986), which assimilate $NH_4^+$ (and amino acid-derived $NH_4^+$) to sustain their glutamate-glutamine cycle. Vice versa, the labile dissolved organic matter derived from bacterial decomposition of refractory organic matter provides a valuable food source for some benthic foraminifera, and is indispensable for the reproduction of some foraminiferal species (Jorissen et al., 1998; Muller and Lee, 1969; Nomaki et al., 2011). In many marine diatoms, which are the main drivers of marine primary productivity, $NH_4^+$ is the preferred source for nitrogen uptake over $NO_3^-$ (Sivasubramanian and Rao, 1988). Foraminifera could act as important nutrient providers for closely associated diatoms, which are also considered as one of their main food sources (Lee et al., 1966). Consequently, the kleptoplast-hosting metabolism in *H. germanica* could benefit from regenerated nitrogen sources by the high OM mineralization rates in *A. tepida*. In summary, foraminiferal carbon and nitrogen fluxes constitute an important link in the food web complex of primary consumers and decomposers.

**Minor comments**

R1: Line 12. Should ‚13C & 15N' be ‚$^{13}$C & $^{15}$N'? This comment applies to the rest of the manuscript.

JW: 13C & 15N were substituted by $^{13}$C and $^{15}$N.

R1: Line 14-19. Throughout the mansuscripte, the results obtained in *A. tepida* are discussed before those obtained in *H. germanica*. I recommend maintaining the same structure in the abstract, as well.

JW: The sequence in the abstract was changed:

Line 13 – 21: "*Ammonia tepida* showed a very high, temperature-influenced intake and turnover rates with more excessive carbon turnover, compared to nitrogen. The quite low metabolic nitrogen turnover in *H. germanica* was not affected by temperature and was higher than the carbon turnover. This might be related with the chloroplast husbandry in *H. germanica* and its lower demands for food derived nitrogen sources. *Ammonia tepida* prefers a soft chlorophyte food source over diatom detritus, which is harder to break down. In conclusion, *A. tepida* shows a generalist behaviour that links with high fluxes of organic matter (OM). Due to its high rates of OM processing and abundances, we conclude that *A. tepida* is an important key-player in intertidal carbon and nitrogen turnover, specifically in the short-term processing of OM and the mediation of dissolved nutrients to associated microbes and primary producers. In contrast, *H. germanica* is a highly specialized species with low rates of carbon and nitrogen budgeting."

R1: Line 25: „Coastal sediments represent the largest pool of marine particulate organic matter (OM)...' Can the authors add some numbers (maybe a percentage?) regarding how big the OM pool is in coastal sediments? In my opinion, such a number will provide a good context to discuss the data obtained from the experiments and to discuss the importance of remineralization processes mediated by benthic foraminifera in coastal environments.

JW: The following sections have been added:

Line 24-31: „Oceanic and terrestrial systems are connected by the carbon cycling in coastal waters, which contribute to a major part of the global carbon cycles and budgets (Bauer et al., 2013; Cai, 2011; Cole et al., 2007; Regnier et al., 2013). Estuaries are an important source for organic matter in coastal systems and were estimated to account for ~ 40% of oceanic phytoplankton primary productivity (Smith and Hollibaugh 1993). Most estuarine areas are considered to be net heterotrophic, or act as carbon sinks, respectively (e.g. Caffrey, 2003, 2004; Cai, 2011; Herrmann et al., 2015). In general, 30% of overall coastal carbon is lost by metabolic oxidation (Smith and Hollibaugh 1993)."

Line 336-341: "As mentioned above, in the heterotrophic, coastal zone 30% of the carbon pool are lost as via respiration. On the other hand, dissolved organic carbon sources from organismic excretion can serve as an important nutrient source for bacteria (e.g., Kahler et al., 1997; Snyder & Hoch, 1996; Zweifel et al., 1993). Therefore, the fast processing of OM in *A. tepida* might be an

important sink for inorganic carbon ($CO_2$ respiration) and at the same time a link for dissolved organic carbon sources in intertidal carbon and nitrogen fluxes."

R1: Line 36: „e.g., temperature or OM quality". This should be „temperature and/or OM quality".

JW: this was changed according to the reviewers suggestion.

R1: Lines 40-41 and 47-48. These sentences are not very clear. Please rephrase.

JW: these sentences were rephrased as follows:

Line 49-53: „Typically, tidal flats offer a high availability of food sources for phytodetrivores or herbivores feeding on microalgae. But dense populations of *A. tepida* communities can deplete sediments from OM sources and consequently control benthic meiofaunal community structures (Chandler, 1989). Therefore, resource partitioning or different metabolic strategies can be beneficial for foraminifera which share the same spatial and temporal habitats."

Line 82-85: "Therefore, seasonal temperature fluctuations and human induced global warming can have a strong impact on foraminiferal community compositions and foraminiferal C & N fluxes."

2 further sentences were added: "In estuaries e.g. temperature acts in many cases as the most controlling factor on metabolic rates and hence on net ecosystem metabolism (Caffrey, 2003). Therefore, this factor was included in one of our observations concerning foraminiferal OM processing."

R1: Lines 58-59. Considering that the experiment described at lines 58-59 is Experiment #2, I suggest moving this sentence after the sentence at lines 60-61, which refers to Experiment #1.

JW: This shift was done:

Line 90-94: „We compared diatom detritus intake and retention of food-derived carbon (pC) and nitrogen (pN) of *A. tepida* and *H. germanica* at three different temperatures (15°C, 20°C, 25°C). The evaluation of the metabolic costs of pC and pN during a 24 hour starvation period can further help to explain species specific OM processing due to metabolic nutrient budgets. Further, both food sources were offered simultaneously to *A. tepida* to identify feeding preferences of this species."

R1: M2 should be $m^2$

JW: replaced with $m^2$

R1: „Individuals were picked from the sediment in sufficient and collected (...)". In sufficient number?

JW: Yes, sentence was completed:

Line 108: „Foraminifera were picked from the sediment in sufficient number and collected (...)".

R1: Line 77: „*Dunaliella tertiolecta* and *Phaeodactylum tricornutum*". The scientific name was already defined at line 58, so this should be *D. tertiolecta* and *P. tricornutum*. This comment applies to the rest of the manuscript, with the exception of tables and figures.

JW: These changes were carried out.

R1: „The experiments started after accumulation of sufficient foraminiferal material three weeks after the field sampling." I assume the authors achieved foraminiferal reproduction during the initial incubation. If my assumption is correct, then it would be good to specify so and provide some information about the conditions used to maintain the foraminifera prior the beginning of the experiments. If the authors know, it might be of interest to know how successful the reproduction event was.

JW: Upon arrival at the lab, the sediment was immediately transferred into aerated aquaria containing filtered seawater at the sampling site. We did not monitor reproduction during the incubation period.

The following sentence was added to the revised manuscript:

Line 107-108: „The sediment samples were kept within aquaria, containing filtered water collected at the sampling site."

R1: Line 84. NaH13CO3, Na15NO3 should be $NaH^{13}CO_3$, $Na^{15}NO_3$.

JW: changed.

R1: Line 88. C.f. should be cf. This comment applies to the rest of the manuscript.

JW: changed.

R1: Line 108. What do the authors mean with „carbon and nitrogen costs of the two species during the period without food"?

JW: sentecne changed: Line 196-197: „...metabolic carbon and nitrogen loss of the two species during the period without food."

R1: Line 114. Cm-2 should be $cm^{-2}$.

JW: changed.

R1: Line 135. A parenthesis is missing.

    JW: Parenthesis added.

R1: Line 137. I suggest including the word „cytoplasm" prior „isotope analysis", for clarity.

    JW: The word „cytoplasm" was included.

R1: Line 153 (formula #2). atomXsample – should this be atom%Xsample? Same for background.

    JW: „atomXsample" was replaced by „atom%Xsample" in both cases.

R1: Line 155. I recommend writing the $I_{iso}$ formula as the other formulas, for clarity.

    JW: The $I_{iso}$ formula was written as the other formulas.

R1: Line 155. There is an extra period after Table 2.

    JW: Extra period removed.

R1: Line 205. No comma needed.

    JW: Comma removed.

R1: Line 212. Phaeodactylum tricornutum should be italic.

    JW: Phaeodactylum tricornutum was changed to „*P. tricornutum*".

R1: Section 4.1 revise references – e.g., a comma is missing between the authors' names and the year of publication and a semicolon should be used to separate different references.

    JW: The reference style was adapted to biogeosciences.

R1: Line 232. Missing parenthesis.

    JW:  Parenthesis added.

R1: Line 250. Almagor et al. – publication year 1981.

    JW: Publication year added.

R1: Lines 281 and 295. Missing parenthesis around the year of publication.

JW: Parenthesis added.

R1: Lines 288 and 295. Comp. should be probabyl cf.

JW: Comp. replaced by cf.

---

## Author Comment (AC2) · 15 Sep 2018

R2: I felt that the rationale to translate these experiments to field based interpretations were rather limited – I suggest the authors strengthen this aspect of the manuscript, making it clear what the findings mean in terms of field-context by reference to a wider literature. If this is not possible, then the translation of these results from laboratory to field studies should be treated with greater caution e.g. tone-down statements such as on line 311.

JW: The following section was added to the manuscript:

Line 315-324: "Our phytodetritus uptake estimates propose, that the foraminiferal biomass consists of $\sim 6 - 8\%$ diatom-derived pC /TOC, with the major amount contained within A. tepida (compare Table 3). An in-situ feeding experiment with deep-sea foraminifera resulted in values of $\sim 1 - 12\%$ pC/TOC (Nomaki et al., 2005b). Similar in-situ incubations in the core of the oxygen minimum zone of the Arabian Sea report $\sim 15\%$ pC/TOC in epifaunal and shallow infaunal foraminiferal carbon uptake (Enge et al., 2014). In-situ incubations offer results closest to the natural responses of organisms in their natural habitat and enable precise estimates of foraminiferal nutrient fluxes. Although, specific microhabitat conditions can have a strong influence on organismic behaviour. The artificial conditions in laboratory experiments also have an influence on physiological analysis, therefore the obtained results should be treated with caution. However, our estimates lie in the same order of magnitude as the above mentioned in-situ studies and offer a basis for estimations on foraminiferal carbon and nitrogen fluxes."

R2: Sections of the manuscript, such as 3.3 are very interesting but take a very linear approach – again, cross-reference to any extended literature might strengthen these arguments.

JW: We hope, that the additional paragraph provided in 3.4 works against the section's former linearity:

Line 323-333: "Kleptoplasty is a wide spread phenomenon in foraminifera, specifically in species inhabiting dysoxic sediments, where kleptoplasts could promote survival in anoxic pore waters (Bernhard et al., 1999). They might be involved in biochemical pathways within the foraminiferal cytoplasm, e.g. the transport of inorganic carbon and nitrogen (LeKieffre et al., 2018). Further, transmission electron microscopic investigations on H. germanica report a very limited abundance of food vesicles (Goldstein et al., 2018). Kleptoplast-bearing species might occupy a distinct niche concerning their

energetic demands. Additionally, they might play a not yet discovered importance in the fluxes of inorganic or dissolved carbon and nitrogen compounds. However, secondary producers with high uptake rates and a quick response to particulate OM sources like A. tepida play a strong role in the biogeochemical carbon and nitrogen recycling.

R2: The discussion leaves the reader with a sense of some "loose ends", so again – perhaps some editing of the discussion to focus on a stronger connection between experiments and field would be helpful. Try to avoid, as in the conclusion (section 5) open-ended discussion where the role for bacteria, for example are never quite tied-down.

JW: Several new sections were added in the discussion (see answers to R1). Further, Line 301-308 were removed in the revised manuscript. We hope this improves the discussion.

R2: Please ensure that you include a proper and complete review of the recent literature (e.g. Jauffrais et al. 2016) on kleptoplasty – you can largely include this in the introduction/state-of-the-art; why not take the opportunity to highlight that "uptake" remains a critical feeding strategy and despite these exciting new developments, the focus of your manuscript illustrates the critical role of benthic Foraminiferal feeding as a key component in the benthic biogeochemical cycle of the intertidal environment – can you say this?

JW: The following section was added to the introduction:

Line 60-78: "A major, important difference between the two species subject to this study is the fact, that H. germanica hosts functional plastids derived from ingested microalgae (Jauffrais et al., 2016; Lopez, 1979), a phenomenon known as klepto-plasty, which was first described for a sacoglossan opisthobranch (Trench, 1969). It was shown, that diatom-derived chloroplasts in the cytoplasm of H. germanica retain their function (as photosynthetically active kleptoplasts) for up to two weeks (Jauffrais et al., 2016). Further, there is recent proof that H. germanica takes up inorganic carbon

and nitrogen sources (HCO3 and NH4+) from the surrounding seawater, most likely to generate metabolites in autotrophic-heterotrophic interactions with its kleptoplasts (LeKieffre et al., 2018). Consequently, the mixotrophic lifestyle of H. germanica might lead to a lower demand of carbon and nitrogen sources and thus to a lower ingestion of various particulate OM sources as food sources. In contrary, food-derived chloroplasts in A. tepida lose their photosynthetic activity after a maximum of 24 hours (Jauffrais et al., 2016)."

R2: Personally, I think you could develop the illustrations/figures – these can be helpful to the readership and I would be tempted to add more, including a location map and some supplementary SEM images of the species – as noted above the genus Ammonia is particularly problematic and displays cryptic diversity, does it not?

JW: A location map was added. A supplementary table containing SEM pictures of the species was added.

[Figure]

**Fig. 1.** Sampling Area

[Figure]

Supplementary Figure 1. a) Light microscope image of fresh picked *A. tepida* specimens (scale bar = 500 μm). b) *A. tepida* after feeding on fresh microalgae. c) Fresh picked *H. germanica* specimens (scale bar = 500 μm). d) *H. germanica* individual (scale bar = 200 μm). e)-h) SEM images of *A. tepida* collected in 2014 at the sampling location of this study (scale bar = 200 μm). i)-j) *H. germanica* collected in 2014 at the sampling location of this study (scale bar = 200 μm). k)-l) *A. tepida* collected in 2016 at the sampling location of this study (scale bar = 200 μm). m) *H. germanica* collected in 2016 at the sampling location of this study (scale bar = 200 μm).

**Fig. 2.** Supplementary Figure

---

## Author Comment (AC3) · 18 Sep 2018

R2: I felt that the rationale to translate these experiments to field based interpretations were rather limited – I suggest the authors strengthen this aspect of the manuscript, making it clear what the findings mean in terms of field-context by reference to a wider literature. If this is not possible, then the translation of these results from laboratory to field studies should be treated with greater caution e.g. tone-down statements such as on line 311.

JW: The following section was added to the manuscript:

Line 315-324: "Our phytodetritus uptake estimates propose, that the foraminiferal biomass consists of $\sim 6-8\%$ diatom-derived pC /TOC, with the major amount contained within A. tepida (compare Table 3). An in-situ feeding experiment with deep-sea foraminifera resulted in values of $\sim 1-12\%$ pC/TOC (Nomaki et al., 2005b). Similar in-situ incubations in the core of the oxygen minimum zone of the Arabian Sea report $\sim 15\%$ pC/TOC in epifaunal and shallow infaunal foraminiferal carbon uptake (Enge et al., 2014). In-situ incubations offer results closest to the natural responses of organisms in their natural habitat and enable precise estimates of foraminiferal nutrient fluxes. Although, specific microhabitat conditions can have a strong influence on organismic behaviour. The artificial conditions in laboratory experiments also have an influence on physiological analysis, therefore the obtained results should be treated with caution. However, our estimates lie in the same order of magnitude as the above mentioned in-situ studies and offer a basis for estimations on foraminiferal carbon and nitrogen fluxes."

R2: Sections of the manuscript, such as 3.3 are very interesting but take a very linear approach – again, cross-reference to any extended literature might strengthen these arguments.

JW: We hope, that the additional paragraph provided in 3.4 works against the section's former linearity:

Line 323-333: "Kleptoplasty is a wide spread phenomenon in foraminifera, specifically in species inhabiting dysoxic sediments, where kleptoplasts could promote survival in anoxic pore waters (Bernhard et al., 1999). They might be involved in biochemical pathways within the foraminiferal cytoplasm, e.g. the transport of inorganic carbon and nitrogen (LeKieffre et al., 2018). Further, transmission electron microscopic investigations on H. germanica report a very limited abundance of food vesicles (Goldstein et al., 2018). Kleptoplast-bearing species might occupy a distinct niche concerning their

energetic demands. Additionally, they might play a not yet discovered importance in the fluxes of inorganic or dissolved carbon and nitrogen compounds. However, secondary producers with high uptake rates and a quick response to particulate OM sources like A. tepida play a strong role in the biogeochemical carbon and nitrogen recycling.

R2: The discussion leaves the reader with a sense of some "loose ends", so again – perhaps some editing of the discussion to focus on a stronger connection between experiments and field would be helpful. Try to avoid, as in the conclusion (section 5) open-ended discussion where the role for bacteria, for example are never quite tied-down.

JW: Several new sections were added in the discussion (see answers to R1). Further, Line 301-308 were removed in the revised manuscript. We hope this improves the discussion.

R2: Please ensure that you include a proper and complete review of the recent literature (e.g. Jauffrais et al. 2016) on kleptoplasty – you can largely include this in the introduction/state-of-the-art; why not take the opportunity to highlight that "uptake" remains a critical feeding strategy and despite these exciting new developments, the focus of your manuscript illustrates the critical role of benthic Foraminiferal feeding as a key component in the benthic biogeochemical cycle of the intertidal environment – can you say this?

JW: The following section was added to the introduction:

Line 60-78: "A major, important difference between the two species subject to this study is the fact, that H. germanica hosts functional plastids derived from ingested microalgae (Jauffrais et al., 2016; Lopez, 1979), a phenomenon known as klepto-plasty, which was first described for a sacoglossan opisthobranch (Trench, 1969). It was shown, that diatom-derived chloroplasts in the cytoplasm of H. germanica retain their function (as photosynthetically active kleptoplasts) for up to two weeks (Jauffrais et al., 2016). Further, there is recent proof that H. germanica takes up inorganic carbon

and nitrogen sources (HCO3 and NH4+) from the surrounding seawater, most likely to generate metabolites in autotrophic-heterotrophic interactions with its kleptoplasts (LeKieffre et al., 2018). Consequently, the mixotrophic lifestyle of H. germanica might lead to a lower demand of carbon and nitrogen sources and thus to a lower ingestion of various particulate OM sources as food sources. In contrary, food-derived chloroplasts in A. tepida lose their photosynthetic activity after a maximum of 24 hours (Jauffrais et al., 2016)."

R2: Personally, I think you could develop the illustrations/figures – these can be helpful to the readership and I would be tempted to add more, including a location map and some supplementary SEM images of the species – as noted above the genus Ammonia is particularly problematic and displays cryptic diversity, does it not?

JW: A location map was added. A supplementary table containing SEM pictures of the species was added.

Please also note the supplement to this comment:
https://www.biogeosciences-discuss.net/bg-2018-231/bg-2018-231-AC3-supplement.pdf

[Figure]

Supplementary Figure 1. a) Light microscope image of fresh picked *A. tepida* specimens (scale bar = 500 µm). b) *A. tepida* after feeding on fresh microalgae. c) Fresh picked *H. germanica* specimens (scale bar = 500 µm). d) *H. germanica* individual (scale bar = 200 µm). e)-h) SEM images of *A. tepida* collected in 2014 at the sampling location of this study (scale bar = 200 µm). i)-j) *H. germanica* collected in 2014 at the sampling location of this study (scale bar = 200 µm). k)-l) *A. tepida* collected in 2016 at the sampling location of this study (scale bar = 200 µm). m) *H. germanica* collected in 2016 at the sampling location of this study (scale bar = 200 µm).

[Figure]

**Fig. 1.**

**Supplement:**

[revised manuscript text omitted]

Supplementary Figure 1. a) Light microscope image of fresh picked *A. tepida* specimens (scale bar = 500 µm). b) *A. tepida* after feeding on fresh microalgae. c) Fresh picked *H. germanica* specimens (scale bar = 500 µm). d) *H. germanica* individual (scale bar = 200 µm). e)-h) SEM images of *A. tepida* collected in 2014 at the sampling location of this study (scale bar = 200 µm). i)-j) *H. germanica* collected in 2014 at the sampling location of this study (scale bar = 200 µm). k)-l) *A. tepida* collected in 2016 at the sampling location of this study (scale bar = 200 µm). m) *H. germanica* collected in 2016 at the sampling location of this study (scale bar = 200 µm).

Table **S1**. Raw data of EA-IRMA of foraminiferal samples of Paper 4 (n.a. = natural abundance, data for 5.5 P from Paper 3).

| | d 15N/14N | AT% 15N/14N | d 13C/12C | AT% 13C/12C |
|---|---|---|---|---|
| *H. germanica* n.a. | 12.58 | 0.371 | -13.56 | 1.091 |
| 5.7 D | 45528.61 | 14.613 | 1153.00 | 2.351 |
| 7.1 D | 216695.58 | 44.464 | 10361.59 | 11.271 |
| 5.5 P | 32011.61 | 15.824 | 1627.25 | 4.389 |

| food source | T | treatment | Nr/Ind | weight [mg] | d 15N/14N | AT% 15N/14N | d 13C/12C | AT% 13C/12C | µg N | µg C |
|---|---|---|---|---|---|---|---|---|---|---|
| 5.7 D | 15°C | 24 hrs fed | 48 | 1.250 | 425.11 | 0.521 | 3.82 | 1.110 | 8.85 | 51.58 |
| 5.7 D | 15°C | 24 hrs fed | 52 | 1.277 | 416.26 | 0.518 | 3.25 | 1.109 | 8.79 | 52.80 |
| 5.7 D | 15°C | 24 hrs fed | 53 | 1.289 | - | - | - | - | - | - |
| 5.7 D | 20°C | 24 hrs fed | 50 | 0.963 | 328.42 | 0.486 | -0.42 | 1.105 | 7.75 | 43.83 |
| 5.7 D | 20°C | 24 hrs fed | 53 | 1.284 | 323.13 | 0.484 | -0.54 | 1.105 | 8.61 | 51.50 |
| 5.7 D | 20°C | 24 hrs fed | 56 | 1.353 | 335.27 | 0.489 | 0.39 | 1.106 | 8.74 | 52.34 |
| 5.7 D | 25°C | 24 hrs fed | 48 | 0.972 | 257.92 | 0.461 | -2.90 | 1.102 | 8.85 | 49.25 |
| 5.7 D | 25°C | 24 hrs fed | 54 | 1.301 | 291.73 | 0.473 | -3.30 | 1.102 | 9.19 | 53.82 |
| 5.7 D | 25°C | 24 hrs fed | 50 | 1.176 | 385.61 | 0.507 | 2.53 | 1.108 | 7.89 | 45.38 |
| 5.7 D | 15°C | 24 hrs starved | 49 | 1.052 | 230.04 | 0.450 | -4.70 | 1.101 | 7.58 | 42.71 |
| 5.7 D | 15°C | 24 hrs starved | 61 | - | - | - | - | - | - | - |
| 5.7 D | 15°C | 24 hrs starved | 53 | 1.192 | 211.71 | 0.444 | -5.63 | 1.099 | 8.70 | 51.81 |
| 5.7 D | 20°C | 24 hrs starved | 48 | 1.108 | 241.56 | 0.455 | -5.04 | 1.100 | 7.44 | 41.11 |
| 5.7 D | 20°C | 24 hrs starved | 45 | 1.099 | 254.99 | 0.459 | -5.81 | 1.099 | 7.96 | 43.16 |
| 5.7 D | 20°C | 24 hrs starved | 44 | 1.202 | 234.45 | 0.452 | -6.98 | 1.098 | 7.39 | 42.01 |
| 5.7 D | 25°C | 24 hrs starved | 49 | 1.217 | 337.63 | 0.490 | -3.98 | 1.101 | 7.70 | 39.16 |
| 5.7 D | 25°C | 24 hrs starved | 39 | 1.057 | 321.86 | 0.484 | -4.15 | 1.101 | 7.18 | 40.77 |
| 5.7 D | 25°C | 24 hrs starved | 43 | 1.235 | 374.66 | 0.503 | -2.93 | 1.102 | 6.66 | 36.67 |
| 7.1 D | 15°C | 24 hrs fed | 55 | 1.411 | 496.43 | 0.547 | 35.23 | 1.144 | 9.48 | 55.00 |
| 7.1 D | 15°C | 24 hrs fed | 52 | 1.158 | 619.09 | 0.592 | 50.35 | 1.161 | 9.60 | 57.15 |
| 7.1 D | 15°C | 24 hrs fed | 53 | 1.417 | 481.34 | 0.542 | 41.18 | 1.151 | 8.80 | 51.30 |
| 7.1 D | 20°C | 24 hrs fed | 49 | 1.373 | 1004.73 | 0.732 | 94.37 | 1.209 | 9.01 | 50.38 |
| 7.1 D | 20°C | 24 hrs fed | 50 | 0.907 | 1148.04 | 0.784 | 113.35 | 1.229 | 7.48 | 42.49 |
| 7.1 D | 20°C | 24 hrs fed | 49 | 1.413 | 1017.81 | 0.737 | 108.25 | 1.224 | 9.49 | 54.69 |
| 7.1 D | 25°C | 24 hrs fed | 51 | 1.243 | 853.93 | 0.677 | 80.93 | 1.194 | 8.90 | 48.30 |
| 7.1 D | 25°C | 24 hrs fed | 48 | 1.195 | 934.13 | 0.706 | 83.27 | 1.197 | 7.60 | 43.78 |
| 7.1 D | 25°C | 24 hrs fed | 53 | 1.243 | 764.11 | 0.645 | 65.33 | 1.177 | 8.23 | 46.30 |
| 7.1 D | 15°C | 24 hrs starved | 50 | 1.128 | 432.69 | 0.524 | 12.73 | 1.120 | 7.28 | 40.52 |
| 7.1 D | 15°C | 24 hrs starved | 46 | 1.166 | 465.91 | 0.536 | 16.65 | 1.124 | 7.27 | 43.98 |
| 7.1 D | 15°C | 24 hrs starved | 59 | 1.251 | 416.39 | 0.518 | 15.44 | 1.123 | 8.80 | 51.35 |
| 7.1 D | 20°C | 24 hrs starved | 49 | 1.160 | 555.79 | 0.569 | 23.69 | 1.132 | 7.88 | 44.51 |
| 7.1 D | 20°C | 24 hrs starved | 48 | 1.209 | 715.38 | 0.627 | 35.48 | 1.144 | 7.74 | 42.10 |
| 7.1 D | 20°C | 24 hrs starved | 50 | 1.280 | 771.33 | 0.647 | 34.82 | 1.144 | 8.32 | 45.05 |
| 7.1 D | 25°C | 24 hrs starved | 53 | 1.335 | 797.43 | 0.657 | 33.54 | 1.142 | 8.08 | 45.90 |
| 7.1 D | 25°C | 24 hrs starved | 48 | 1.144 | 855.89 | 0.678 | 51.27 | 1.162 | 7.87 | 41.64 |
| 7.1 D | 25°C | 24 hrs starved | 54 | 1.230 | 845.19 | 0.674 | 43.79 | 1.154 | 8.00 | 45.05 |
| 5.5 P | 15°C | 24 hrs fed | 50 | 1.469 | 541.16 | 0.564 | 64.97 | 1.177 | 8.27 | 49.43 |
| 5.5 P | 15°C | 24 hrs fed | 51 | 1.359 | 618.81 | 0.592 | 68.13 | 1.180 | 9.73 | 56.12 |
| 5.5 P | 15°C | 24 hrs fed | 51 | 1.398 | 559.13 | 0.570 | 60.14 | 1.171 | 9.53 | 54.99 |
| 5.5 P | 20°C | 24 hrs fed | 52 | 1.172 | 598.98 | 0.585 | 58.26 | 1.169 | 10.87 | 60.65 |
| 5.5 P | 20°C | 24 hrs fed | 52 | 1.363 | 576.87 | 0.577 | 62.15 | 1.174 | 10.05 | 52.00 |
| 5.5 P | 20°C | 24 hrs fed | 51 | 1.117 | 484.73 | 0.543 | 57.47 | 1.168 | 10.69 | 53.97 |
| 5.5 P | 25°C | 24 hrs fed | 47 | 1.012 | 439.52 | 0.527 | 32.98 | 1.142 | 9.38 | 46.52 |
| 5.5 P | 25°C | 24 hrs fed | 49 | 1.114 | 461.60 | 0.535 | 33.99 | 1.143 | 9.49 | 46.77 |
| 5.5 P | 25°C | 24 hrs fed | 49 | 1.127 | 403.72 | 0.514 | 34.21 | 1.143 | 10.93 | 54.56 |
| 5.5 P | 15°C | 24 hrs starved | 48 | 1.246 | 364.87 | 0.500 | 44.50 | 1.154 | 7.43 | 40.48 |
| 5.5 P | 15°C | 24 hrs starved | 52 | 1.531 | 397.28 | 0.511 | 47.21 | 1.157 | 9.18 | 52.64 |
| 5.5 P | 15°C | 24 hrs starved | 53 | 1.508 | 351.37 | 0.495 | 48.17 | 1.158 | 9.13 | 50.88 |
| 5.5 P | 20°C | 24 hrs starved | 51 | 1.098 | 395.17 | 0.511 | 43.51 | 1.153 | 10.00 | 49.00 |
| 5.5 P | 20°C | 24 hrs starved | 54 | 1.019 | 365.61 | 0.500 | 40.76 | 1.150 | 9.27 | 45.91 |
| 5.5 P | 20°C | 24 hrs starved | 55 | 1.322 | 362.58 | 0.499 | 31.43 | 1.140 | 9.90 | 48.50 |
| 5.5 P | 25°C | 24 hrs starved | 54 | 1.046 | 289.12 | 0.472 | 18.97 | 1.126 | 9.62 | 50.96 |
| 5.5 P | 25°C | 24 hrs starved | 39 | 0.952 | 333.26 | 0.488 | 22.21 | 1.130 | 7.14 | 36.69 |
| 5.5 P | 25°C | 24 hrs starved | 57 | 1.327 | 313.36 | 0.481 | 19.12 | 1.127 | 10.97 | 52.50 |

Table **S2**. Raw data of GC-IRMS of water samples.

| | food source | T | | avg d13C/12C | avg AT% 13C/12C | ppm CO2 |
|---|---|---|---|---|---|---|
| $H_3PO_4$ | | - | - | -18.734 | 1.0852 | 83.85 |
| $H_3PO_4$ + SW | | - | - | -1.537 | 1.1040 | 9162.91 |
| | 5.7 D | 15°C | | -1.135 | 1.1044 | 9398.27 |
| | 5.7 D | 15°C | | -1.250 | 1.1043 | 8406.95 |
| | 5.7 D | 15°C | | -1.156 | 1.1044 | 9883.24 |
| | 5.7 D | 20°C | | -0.870 | 1.1047 | 9184.63 |
| | 5.7 D | 20°C | | -1.162 | 1.1044 | 7778.55 |
| | 5.7 D | 20°C | | -0.994 | 1.1046 | 8489.17 |
| | 5.7 D | 25°C | | -0.963 | 1.1046 | 8829.57 |
| | 5.7 D | 25°C | | -1.047 | 1.1045 | 9614.38 |
| | 7.1 D | 15°C | | -0.887 | 1.1047 | 9472.19 |
| | 7.1 D | 15°C | | -0.756 | 1.1048 | 9247.92 |
| | 7.1 D | 15°C | | -1.060 | 1.1045 | 9794.93 |
| | 7.1 D | 20°C | | -0.941 | 1.1046 | 6902.25 |
| | 7.1 D | 20°C | | -0.762 | 1.1048 | 7843.66 |
| | 7.1 D | 20°C | | -0.910 | 1.1047 | 7959.04 |
| | 7.1 D | 25°C | | -0.306 | 1.1053 | 7818.50 |
| | 7.1 D | 25°C | | -0.465 | 1.1052 | 9206.42 |
| | 5.7 P | 15°C | | -0.861 | 1.1047 | 7659.74 |
| | 5.7 P | 15°C | | -0.971 | 1.1046 | 9208.57 |
| | 5.7 P | 15°C | | -0.655 | 1.1049 | 9055.14 |
| | 5.7 P | 20°C | | -0.536 | 1.1051 | 9910.84 |
| | 5.7 P | 20°C | | -0.538 | 1.1051 | 8473.16 |
| | 5.7 P | 20°C | | -0.691 | 1.1049 | 9219.05 |
| | 5.7 P | 25°C | | -0.351 | 1.1053 | 7268.07 |
| | 5.7 P | 25°C | | -0.664 | 1.1049 | 7246.05 |
| He | | | | | | 0 |

Response to Referee 2

R2: I felt that the rationale to translate these experiments to field based interpretations were rather limited – I suggest the authors strengthen this aspect of the manuscript, making it clear what the findings mean in terms of field-context by reference to a wider literature. If this is not possible, then the translation of these results from laboratory to field studies should be treated with greater caution e.g. tone-down statements such as on line 311.

> JW: The following section was added to the manuscript:
>
> Line 315-324: "Our phytodetritus uptake estimates propose, that the foraminiferal biomass consists of ~ 6 – 8% diatom-derived pC /TOC, with the major amount contained within *A. tepida* (compare Table 3). An *in-situ* feeding experiment with deep-sea foraminifera resulted in values of ~ 1 – 12% pC/TOC (Nomaki et al., 2005b). Similar *in-situ* incubations in the core of the oxygen minimum zone of the Arabian Sea report ~ 15% pC/TOC in epifaunal and shallow infaunal foraminiferal carbon uptake (Enge et al., 2014). *In-situ* incubations offer results closest to the natural responses of organisms in their natural habitat and enable precise estimates of foraminiferal nutrient fluxes. Although, specific microhabitat conditions can have a strong influence on organismic behaviour. The artificial conditions in laboratory experiments also have an influence on physiological analysis, therefore the obtained results should be treated with caution. However, our estimates lie in the same order of magnitude as the above mentioned *in-situ* studies and offer a basis for estimations on foraminiferal carbon and nitrogen fluxes."

R2: Sections of the manuscript, such as 3.3 are very interesting but take a very linear approach – again, cross-reference to any extended literature might strengthen these arguments.

> JW: We hope, that the additional paragraph provided in 3.4 works against the section's former linearity:
>
> Line 323-333: „Kleptoplasty is a wide spread phenomenon in foraminifera, specifically in species inhabiting dysoxic sediments, where kleptoplasts could promote survival in anoxic pore waters (Bernhard et al., 1999). They might be involved in biochemical pathways within the foraminiferal cytoplasm, e.g. the transport of inorganic carbon and nitrogen (LeKieffre et al., 2018). Further, transmission electron microscopic investigations on *H. germanica* report a very limited abundance of food vesicles (Goldstein et al., 2018). Kleptoplast-bearing species might occupy a distinct niche concerning their energetic demands. Additionally, they might play a not yet discovered importance in the fluxes of inorganic or dissolved carbon and nitrogen compounds. However, secondary producers with high uptake rates and a quick response to particulate OM sources like *A. tepida* play a strong role in the biogeochemical carbon and nitrogen recycling.

R2: The discussion leaves the reader with a sense of some „loose ends", so again – perhaps some editing of the discussion to focus on a stronger connection between experiments and field would be helpful. Try to avoid, as in the conclusion (section 5) open-ended discussion where the role for bacteria, for example are never quite tied-down.

> JW: Several new sections were added in the discussion (see answers to R1). Further, Line 301-308 were removed in the revised manuscript. We hope this improves the discussion.

R2: Please ensure that you include a proper and complete review of the recent literature (e.g. Jauffrais et al. 2016) on kleptoplasty – you can largely include this in the introduction/state-of-the-art; why not take the opportunity to highlight that „uptake" remains a critical feeding strategy and despite these exciting new developments, the focus of your manuscript illustrates the critical role of benthic Foraminiferal feeding as a key component in the benthic biogeochemical cycle of the intertidal environment – can you say this?

JW: The following section was added to the introduction:

Line 60-78: „A major, important difference between the two species subject to this study is the fact, that *H. germanica* hosts functional plastids derived from ingested microalgae (Jauffrais et al., 2016; Lopez, 1979), a phenomenon known as kleptoplasty, which was first described for a sacoglossan opisthobranch (Trench, 1969). It was shown, that diatom-derived chloroplasts in the cytoplasm of *H. germanica* retain their function (as photosynthetically active kleptoplasts) for up to two weeks (Jauffrais et al., 2016). Further, there is recent proof that *H. germanica* takes up inorganic carbon and nitrogen sources ($HCO_3$ and $NH_4^+$) from the surrounding seawater, most likely to generate metabolites in autotrophic-heterotrophic interactions with its kleptoplasts (LeKieffre et al., 2018). Consequently, the mixotrophic lifestyle of *H. germanica* might lead to a lower demand of carbon and nitrogen sources and thus to a lower ingestion of various particulate OM sources as food sources. In contrary, food-derived chloroplasts in *A. tepida* lose their photosynthetic activity after a maximum of 24 hours (Jauffrais et al., 2016)."

R2: Personally, I think you could develop the illustrations/figures – these can be helpful to the readership and I would be tempted to add more, including a location map and some supplementary SEM images of the species – as noted above the genus Ammonia is particularly problematic and displays cryptic diversity, does it not?

JW: A location map was added. A supplementary table containing SEM pictures of the species was added.